# Encapsulation mechanisms and structural studies of GRM2 bacterial microcompartment particles

Gints Kalnins [1]*, Eva-Emilija Cesle[1], Juris Jansons[1], Janis Liepins [2], Anatolij Filimonenko[3] & Kaspars Tars[1,4]

Bacterial microcompartments (BMCs) are prokaryotic organelles consisting of a protein shell and an encapsulated enzymatic core. BMCs are involved in several biochemical processes, such as choline, glycerol and ethanolamine degradation and carbon fixation. Since non-native enzymes can also be encapsulated in BMCs, an improved understanding of BMC shell assembly and encapsulation processes could be useful for synthetic biology applications. Here we report the isolation and recombinant expression of BMC structural genes from the *Klebsiella pneumoniae* GRM2 locus, the investigation of mechanisms behind encapsulation of the core enzymes, and the characterization of shell particles by cryo-EM. We conclude that the enzymatic core is encapsulated in a hierarchical manner and that the CutC choline lyase may play a secondary role as an adaptor protein. We also present a cryo-EM structure of a $pT = 4$ quasi-symmetric icosahedral shell particle at 3.3 Å resolution, and demonstrate variability among the minor shell forms.

[1] Latvian Biomedical Research and Study Centre, Ratsupites 1, Riga 1067, Latvia. [2] Institute of Microbiology and Biotechnology, University of Latvia, Jelgavas 1, Riga 1004, Latvia. [3] Central European Institute of Technology, Masaryk University, Kamenice 753/5, 62500 Brno, Czech Republic. [4] University of Latvia, Jelgavas 1, Riga 1004, Latvia. *email: gints@biomed.lu.lv

Bacterial microcompartments (BMCs) are organelle-like structures consisting of a large quasi-icosahedral or polyhedral protein shell up to 200–300 nm in size and an encapsulated enzymatic core[1–6]. It is estimated that approximately 25% of bacterial taxa have genomes that contain a BMC locus of some sort that performs various functions[7–9]. The most well-studied and well-known types of BMCs are the carboxysomes because they are widespread among cyanobacteria and play an important role in carbon fixation[5,10,11]. Another, more diverse group of BMCs are the metabolosomes. Metabolosomes are specialized BMCs that breakdown various compounds, such as propanediol[12,13], ethanolamine[14], choline[15], and rhamnose/fucose[16,17]. In metabolosomes, the catabolic breakdown of the substrate mostly involves an aldehyde intermediate, which may be toxic to the cells and/or volatile and hence is sequestered within the BMC protein shell[18,19]. Encapsulation of the enzymatic pathway also has the benefit of increasing the local substrate concentrations and increasing the overall efficiency of the pathway. The metabolosomes generally contain at least four or five different encapsulated core enzymes, including signature enzymes that perform the initial breakdown of the substrate, alcohol dehydrogenase, aldehyde dehydrogenase, and phosphotransacylase[5,20].

The BMC shell consists of three types of BMC shell proteins: the BMC-H, BMC-T, and BMC-P proteins[5,6]. BMC-H is a hexameric BMC protein consisting of one Pfam00936 domain, and it is capable of forming uniform flat sheets[2,21–24]. The Pfam00936 domain consists of an α–β double sandwich that contains a four-stranded antiparallel beta sheet and is flanked by two α-helices on one side. The BMC-T protein is somewhat similar to BMC-H according to its sequence, but instead of one Pfam00936 domain, as in BMC-H, it contains two fused domains and as a consequence is trimeric instead of hexameric[25–28]. BMC-H and BMC-T hexamers and trimers have large pores that can be up to 14 Å in size in their centers that are thought to ensure metabolite flow across the BMC shell[19,29,30]. The BMC-P (or BMV)[8,31,32] monomer consists of a single Pfam03319 domain, which is structurally unrelated to the Pfam00936 domain and contains a 5-stranded β-barrel. BMC-P proteins are pentameric, and their function is thought to be limited to capping the vertices of icosahedrons[28,32]. In the formation of BMCs, a key role is played by encapsulation peptide (EP) sequences[33–36]. EPs are small, 10–20 residue-long amphipathic α-helices that are attached to the core enzymes N- or C-terminally or inside the flexible surface loops. Genetic fusions of such sequences have been demonstrated to be sufficient for the encapsulation of non-native proteins in propanediol utilization (Pdu)[35,37], ethanolamine utilization (Eut)[38], and beta-carboxysome[39] BMC shell particles. BMC-T and BMC-H proteins, which are encoded in the Pdu locus[40,41], and a BMC-H protein, which is encoded in the Eut locus[38], have been demonstrated to serve as EP targets, so it seems that the specific shell partner may vary among different types of BMCs. In addition to binding the core to the shell, EPs are thought to have a crosslinking influence in the enzymatic core as well[5,33,42–44].

The glycyl-radical associated microcompartment group 2 (GRM2) type BMC locus encodes the glycyl-radical enzyme-associated microcompartment (GRM) subgroup that includes the choline utilizing locus C (CutC) choline lyase as its signature enzyme[8,9,20]. CutC cleaves its initial substrate, choline, into trimethylamine (TMA) and acetaldehyde[45,46]. TMA itself has been under intense scrutiny as a bacterial metabolite with potential therapeutic importance since its oxidized form, trimethylamine-N-oxide (TMAO), has been identified as a likely contributor to cardiovascular diseases[47,48]. Like most BMC loci, the GRM2 locus contains regulatory, shell, core enzyme, and transporter genes. There are two unusual traits of the GRM2 locus. First, there is a unique, ~340 residue-long N-terminal extension of CutC, which is somewhat homologous to the subsequent 340 residues of CutC and probably originated as an N-terminal duplication[20,46]. The exact function of this extension is unknown, but its involvement in core multimerization or encapsulation processes has been proposed earlier[20]. Another unusual trait of GRM2 is the lack of BMC-T genes in the locus, which is a trait shared with only a few other BMC loci[8,9]. The GRM2 locus therefore contains five structural shell genes encoding four BMC-H proteins and one BMC-P protein.

The versatility of BMCs and their capacity to encapsulate large cargos of various sizes have made them appealing targets for synthetic biology applications. These organelles are generated with the specific goal of packing entire enzymatic pathways to increase their efficiency and lower the effects of intermediate toxicity, making them attractive platforms for the construction of recombinant metabolic pathways. There have been some successful examples, including a two-component ethanol-producing[35,49] BMC-based recombinant particle and a polyphosphate-accumulating[50] compartments. A major obstacle for more extensive research in this area is the lack of a robust encapsulation system. An ideal platform must be universal enough to successfully encapsulate various enzymes of different sizes, solubility, and oligomerization states. Native EPs have been successfully used for this purpose[34,35,37–39,48–50]; however, since they are amphipathic, insolubility or excessive aggregation is a potential outcome. These limits have been circumvented in some recent cases by creating artificial encapsulation mechanisms, either by introducing non-native binding partners in target and shell proteins[51,52] or postponing assembly[53,54]. There are several BMC types that have not yet been studied in detail, and those could reveal themselves as universal and robust biotechnological platforms.

In this study, we demonstrate a practical production system for recombinant GRM2 shell particles from *Klebsiella pneumoniae* and the requirements for recombinant shell formation. We present a 3.3 Å resolution cryo-EM structure of pT = 4 BMC particle, demonstrate the presence of variable minor shell types, and identify the potential roles of the particular core enzymes in the core encapsulation process.

## Results

**Formation of shell particles and effects of BMC-H variants.** In numerous reported cases, the recombinant expression of structural BMC shell genes has successfully yielded stable empty shell particles without enzymatic cores[28,38,39,53]. We extensively investigated what kind of minimal gene set is essential for *Klebsiella pneumoniae* GRM2 shell particle formation (Fig. 1, Table 1, Supplementary Figs. 1–3). We were able to obtain particles which we designated BMC shell-derived particles (BDPs) due to the fact that these are only partially representative of native BMCs—they lack full enzymatic core, are produced in a non-native expression system, and are smaller and more regular than native BMCs. In all cases we purposefully purified BDPs from equivalent amounts of biomass for the results to be comparable. We observed that the minimal requirement for BDP formation is the cmcC + D protein pair, which forms predominantly small type BDPs eluting between 90 and 105 ml on Superose 6 column (Table 1, Supplementary Fig. 3). Curiously, neither for cmcA + D nor cmcB + D we were able to observe formation of BDPs, despite the high similarity between cmcA, cmcB, and cmcC (Fig. 1a). We reasoned that cmcABC should probably be co-expressed from one promoter since these genes in the *Klebsiella pneumoniae* genome are separated by only 8–10 bp-long sequences. Such a construct (cmcABC + D) also resulted predominantly in small type BDPs,

although the yield was much lower than that of the cmcC + D variant. Remarkably, cmcAB + D construct was able to form low amounts of small type particles despite cmcA + D and cmcB + D unable to do so (Supplementary Fig. 3). It is possible that some kind of a synergistic effect between cmcA and cmcB is responsible for this ability to form BDPs.

Due to a defective oligonucleotide used in the PCR, we accidentally created a mutant cmcC variant, designated cmcC′, containing a frameshift mutation that affected the last five cmcC residues and created an additional elongation containing eight residues (Fig. 1b). We observed that the yield of the cmcABC′ + D BDPs was greatly increased when compared to that of cmcABC + D (Fig. 2a, Supplementary Fig. 1b). Based on the sodium dodecyl sulfate–polyacrylamide gel electrophoresis (SDS-PAGE) analysis of gel filtration fractions (Fig. 2a), we also concluded that cmcABC′ + D particles are not uniform in size, as some eluted immediately after the void volume of 60 ml (designated large type particles), while some material formed a discrete peak at 90–100 ml (designated small type particles), and the rest were spread out in the intermediate zone between these two peaks (designated intermediate particles). Negative staining transmission electron microscopy (TEM) analysis of the particles

in these zones (Fig. 2b, Supplementary Figs. 4–7) confirmed that purified cmcABC′ + D BDPs are indeed different in size and are partially sorted during gel filtration according to their sizes; the 90–100 ml peak contained predominantly 20–30 nm particles, and the BDP size increased in the direction of the large type particle peak, which contained particles up to 200 nm in size. This confirmed that the wide distribution of the shell proteins in the gel filtration is a result of the presence of different sizes of particles and not merely because of the aggregation of small particles. Surprisingly, truncation of cmcC in cmcABC_trunc + D also resulted in the formation of large type BDPs (Table 1, Supplementary Fig. 2a), while this truncation did not have any observable effect on the formation of small type particles in cmcC_trunc + D BDPs (Supplementary Fig. 3b).

In our subsequent experiments, we co-expressed the full structural gene sets cmcABC + D + E and cmcABC′ + D + E. cmcE contained an around 30 amino acid C-terminal elongation if compared to cmcA, cmcB, or cmcC (Supplementary Fig. 8), and we observed its co-migration with the BDPs (Supplementary Fig. 1d, e). While cmcE had no influence on the size distribution or yield of the cmcABC′ + D BDPs, for the native gene set cmcABC + D cmcE inclusion caused the formation of the large type particles in a pattern similar to that of cmcABC′ + D (Supplementary Fig. 1d). We also tested both mutant and C-terminally truncated cmcC′ + D and cmcC_trunc + D BDP variants (Fig. 1b, Supplementary Figs. 3a, b) and observed that there were no differences in the yield or size distribution of BDPs between all three cmcC variants (Supplementary Fig. 3). An obvious common trait of cmcE and mutant cmcC′ is the presence of a C-terminal elongation that consists of 8 residues for cmcC′ and 40 residues for cmcE.

When the cmcABC and cmcABC′ were expressed in the absence of BMC-P cmcD, we could purify almost none of the BMC proteins in the case of cmcABC; however, for cmcABC′, it was possible to observe an even "smear" of BMC proteins at the gel filtration volumes of 60–100 ml (Supplementary Fig. 1c). A very similar "smear" was also observed for cmcE particles during cmcE + D co-expression (Supplementary Fig. 3f). In this latter case, cmcD was not observed in the gel filtration profile, suggesting a lack of interaction between these two proteins. When analyzing such "smeared" material of the cmcABC gel filtration 40–42 ml and cmcE + D 94–96 ml fractions with TEM (Supplementary Fig. 9), both rounded and elongated nanotube-like particles could be observed. Curiously, although both were C-terminally elongated, cmcC′ and cmcE had different abilities to

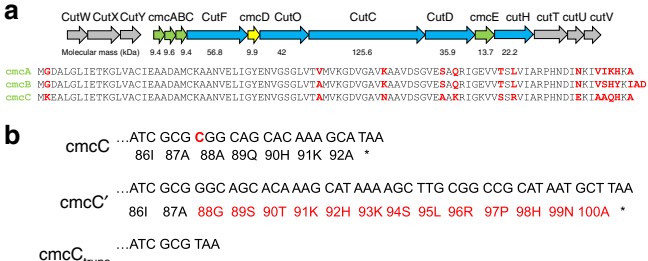

**Fig. 1 Klebsiella pneumoniae GRM2 locus and variants of cmcC. a** Klebsiella pneumonia GRM2 locus. Structural shell BMC-H proteins cmcA, cmcB, cmcC, and cmcE are colored in green, and BMC-P protein cmcD is colored in yellow. Core enzymes CutF (aldehyde dehydrogenase), CutO (alcohol dehydrogenase), CutC (choline lyase), CutD (glycyl-radical activating enzyme), and CutH (phosphotransacylase) are colored in blue. Regulatory and transporter genes are colored in gray. The genes have been named according to previous research[15]. **b** C-terminal amino acid sequences of three cmcC variants—cmcC (native), cmcC′ (mutated), and cmcC_trunc (truncated).

**Table 1 Summary of BDP self-assembly experiments (Supplementary Figs. 1–3 and 12).**

| pET-Duet-1 T7-1 | pET-Duet-1 T7-2 | pRSF-Duet T7-1 | Results |
|---|---|---|---|
| cmcABC | – | – | No purified particles |
| cmcABC | cmcD | – | Predominantly small type particles |
| cmcABC | cmcD | cmcE | Large, intermediary, and small type particles |
| cmcABC′ | cmcD | cmcC′ | Large, intermediary, and small type particles |
| cmcABC′ | cmcD | cmcAB | Large, intermediary, and small type particles |
| cmcABC′ | – | – | Irregular "smeared" material |
| cmcABC′ | cmcD | – | Large, intermediary, and small type particles |
| cmcABC′ | cmcD | cmcE | Large, intermediary, and small type particles |
| cmcABC_trunc | cmcD | – | Large, intermediary, and small type particles |
| cmcAB | cmcD | – | Predominantly small type particles |
| cmcA | cmcD | – | No purified particles |
| cmcB | cmcD | – | No purified particles |
| cmcC | cmcD | – | Predominantly small type particles |
| cmcC | cmcD | cmcE | Predominantly small type particles |
| cmcE | cmcD | – | Irregular "smeared" material |
| cmcC′ | cmcD | – | Predominantly small type particles |
| cmcC_trunc | cmcD | – | Predominantly small type particles |

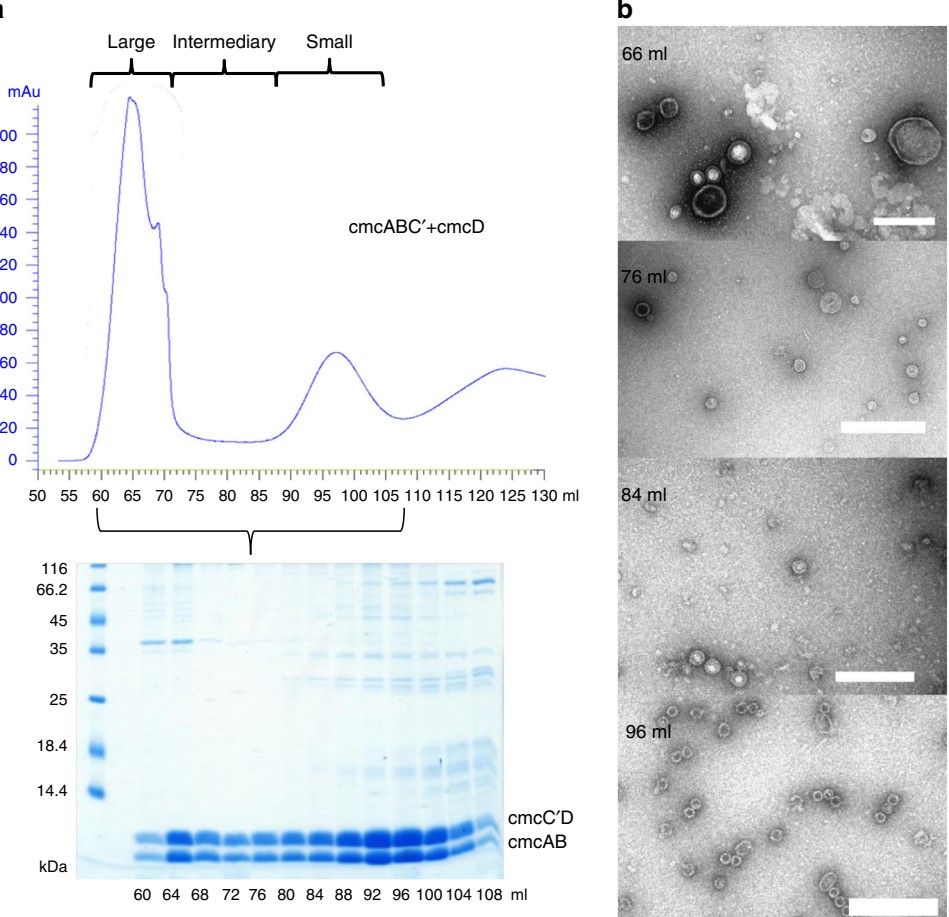

**Fig. 2 Characterization of cmcABC′ + cmcD BDPs. a** Gel filtration of sedimented BDPs and SDS-PAGE analysis of the fractions. Two noticeable BDP peaks were formed: one immediately after the empty volume of 60 ml (large type particle zone) and one at approximately 90–100 ml (small type particle zone). An intermediary zone with smaller BDP protein amounts was formed between these two zones. **b** Examples of TEM analysis of the BDP samples in the large particle zone (66 ml), intermediary zone (76 and 84 ml), and small particle zone (96 ml). Scale bar: 200 nm.

interact with the pentameric units; the former is able to interact with them, while the latter is unable to interact with them.

Mass spectrometry analysis identified peaks matching all four shell proteins in the cmcABC + D, cmcABC′ + D and cmcABC$_{trunc}$ + D BDPs (Supplementary Fig. 10a–e). cmcC′ was identified as the major BMC-H protein in cmcABC′ + D BDPs. Curiously, in the case of cmcABC$_{trunc}$ + D BDPs the major protein was cmcB. It was virtually impossible to distinguish between cmcB and cmcC in cmcABC + D small type BDPs, so both could be the major shell components. While cmcD could be detected in small type particles, it was practically undetectable in large type particles (Supplementary Fig. 10a, b and 10d, e). This would be expected, since increased triangulation numbers in larger particles would significantly reduce the proportion of fivefold vertices occupied by cmcD. In cmcABC′ + D particles the proportion of BMC-H proteins among large and small type BDPs were largely similar; however, the proportion of cmcC$_{trunc}$ was significantly increased in large type cmcABC$_{trunc}$ + D BDPs (Supplementary Fig. 10d, e). Unlike cmcC′, cmcC$_{trunc}$ is not the major protein in cmcABC$_{trunc}$ + D BDPs but instead is present in minor amounts in large type particles. Also, we identified several unexpected $m/z$ peaks in several cases (Supplementary Fig. 10a, b, d, e, i). The identities of these peaks could be degradation products of BMC proteins. Since we did not observed any such peaks in cmcC′ + D, cmcC$_{trunc}$ + D, or cmcC + D small type BDPs (Supplementary Fig. 10f–h), and they appear in cmcAB + D particles (Supplementary Fig. 10i), source of these peaks are cmcA

and/or cmcB proteins. It is unclear whether these products have any significant impact on assembly process.

We compared expression levels of BMC-H components of cmcAD, cmcBD, cmcCD, cmcC′D, and cmcC$_{trunc}$D constructs and observed in SDS-PAGE gel that the expression levels of these proteins are fairly similar (Supplementary Fig. 11), with cmcC$_{trunc}$ exhibiting slightly lower expression levels than others. It could be possible that these lower expression levels are responsible for its low content in cmcABC$_{trunc}$ + D BDPs. All tested proteins were also soluble in similar amounts, except in the case of cmcA. cmcA is much more insoluble than other BMC-H proteins, although mass spectrometry analysis confirmed its inclusion in soluble composite cmcABC′ + D, cmcABC + D, and cmcABC$_{trunc}$ + D BDPs (Supplementary Fig. 10a-b and c-d). Thus, the solubility and availability of BMC-H proteins could be dependable on the composition of other shell components. We tested the influence of expressing additional cmcC′and cmcAB from another promoter on the BDP size and yield (Supplementary Fig. 12); however, we did not observe any dramatic differences in yield or particle size distributions in gel filtration, just a very minor improvement of overall yield in case of additional cmcC′. It is possible that additional cmcC′ or cmcAB expressed from a different promoter do not take part in the complex interplay between cmcA, cmcB, and cmcC genes located so closely to each other during their translation; perhaps formation of particular heterohexamers consisting of more than one type of BMC-H protein is the key to this process.

**Table 2 Summary of BDP encapsulation experiments (Supplementary Figs. 13–16).**

| pET-Duet-1 T7-1 | pET-Duet-1 T7-2 | pRSF-Duet-1 T7-1$_1$ | pRSF-Duet-1 T7-2$_1$ | pRSF-Duet-1 T7-1$_2$ | pRSF-Duet-1 T7-2$_2$ | Results |
|---|---|---|---|---|---|---|
| cmcABC' | cmcD | CutC | – | – | – | CutC in all zones |
| cmcABC' | cmcD | CutF | – | – | – | No encapsulation observed |
| cmcABC' | cmcD | CutO | – | – | – | No encapsulation observed |
| cmcABC' | cmcD | CutH | – | – | – | No encapsulation observed |
| cmcABC' | cmcD | CutC | cmcE | – | – | CutC in all zones |
| cmcABC' | cmcD | CutF | cmcE | – | – | No encapsulation observed |
| cmcABC' | cmcD | CutO | cmcE | – | – | No encapsulation observed |
| cmcABC' | cmcD | CutH | cmcE | – | – | No encapsulation observed |
| cmcABC' | cmcD | CutC$_{1–325}$ | – | – | – | No encapsulation observed |
| cmcABC' | cmcD | CutC$_{336–1128}$ | – | – | – | CutC$_{336–1128}$ in all zones |
| cmcABC' | cmcD | CutC$_{336–1128}$ | CutO | – | – | CutO and CutC$_{336–1128}$ in all zones |
| cmcABC' | cmcD | CutC$_{336–1128}$ | CutF | – | – | CutC$_{336–1128}$ in all zones |
| cmcABC' | cmcD | CutC | CutO | – | – | CutO and CutC in all zones |
| cmcABC' | cmcD | CutC | CutF | – | – | CutC in all zones, CutF predominantly in large type particle zone |
| cmcABC' | cmcD | CutC | CutH | – | – | CutC in all zones |
| cmcABC' | cmcD | CutC | CutF | CutO | – | CutC, CutF, and CutO predominantly in large type particle zone |
| cmcABC' | cmcD | CutC | CutF | CutO | CutH | CutC, CutF, and CutO predominantly in large type particle zone |
| cmcC' | cmcD | CutC | – | – | – | CutC predominantly in small type particle zone |
| cmcC' | cmcD | CutC | CutF | CutO | – | No encapsulation observed |
| cmcC | cmcD | CutC | – | – | – | CutC predominantly in small type particle zone |
| cmcC | cmcD | CutC | CutF | CutO | – | No encapsulation observed |

**Hierarchy of the GRM2 core encapsulation mechanism**. Since the GRM2 locus has several unusual traits, we wanted to test whether it is possible to encapsulate some of the core enzymes by the recombinant co-expression of our BDPs. We demonstrate our experiments with three core enzymes: CutC (signature enzyme choline lyase), CutO (alcohol dehydrogenase), and CutF (aldehyde dehydrogenase). CutD (CutC-activating enzyme) is an insoluble protein when produced recombinantly[46], and we were not able to detect encapsulation of CutH (phosphotransacylase) at all in any of tested BDPs (Supplementary Fig. 14c–e). We selected the cmcABC' + D pET-Duet1 construct as the BDP platform for encapsulation experiments since it offered a more convenient co-expression setup in a two-plasmid expression system and a greater yield than that of native cmcABC + D particles. Our results are summarized in Table 2 and Supplementary Figs. 13–16.

To illuminate the role of the unique N-terminal extension of CutC, we created two new constructs by cutting CutC into two parts containing either the N-terminal 326 amino acids or the C-terminal 792 amino acids, which were visible within the electron density of our previously reported crystal structure[46]. We performed control experiments to test whether Superose 6 gel filtration can properly separate core enzymes from BDPs (Supplementary Figs. 17 and 18). For these experiments we used purified His6x-tagged core enzymes and as a size marker representing BDPs we chose 29 nm diameter bacteriophage AP 205 virus-like particles[55]. We concluded that gel filtration on the Superose 6 column is able to separate CutC (116–132 ml), CutO (116–140 ml), and CutH (120–144 ml) from AP 205 virus-like particles migrating as small type BDPs (Supplementary Fig. 11b). This confirmed that these three core enzymes as such generally do not overlap with BDP elution fractions. However, CutF eluted in a pattern both consistent with a tetrameric oligomerization state (roughly the same as a dimeric CutC) and also as a larger complex starting as early as 64 ml (Supplementary Figs. 17a, c and 18a), thus partially overlapping with BDP elution zones. The presence of CutF caused shift of elution of full-length CutC towards larger-sized aggregates as early as 60 ml, but not N-terminally truncated CutC$_{336–1128}$, indicating a crosslinking influence of CutF on full-length CutC. We also tested whether there is some direct interaction between formed shells and core enzymes or any such interactions mediated by *Escherichia coli* proteins. We expressed core enzymes (CutC + CutO or CutC + CutF + CutO) and BDPs (cmcABC' + D) in separate batches, then mixed the biomasses in equivalent amounts, and proceeded further with cell lysis, ultracentrifugation, and gel filtration as usually (Supplementary Figs. 18c, e). CutC + CutO and CutC + CutF + CutO biomasses without addition of cmcABC' + D BDPs were treated in a similar fashion as controls. Only in the case of CutC + CutF + CutO mixed with cmcABC' + D biomass we were able to detect low amounts of CutC core enzyme (Supplementary Fig. 12e), suggesting some association with BDPs. However, neither CutC nor CutO without BDPs, nor CutC or CutO, when mixed with cmcABC' + D BDPs, could be observed in fractions before 108 ml (Supplementary Figs. 12b–d), confirming the absence of association with already formed BDPs. This strongly suggests that the observed co-migration of CutC and CutO with BDPs after co-expression described below is almost certainly due to encapsulation. In gel filtration on Superose 6 column CutO is generally eluted slightly later than CutC (Supplementary Fig. 17) and in 6xHis tagged CutC pull-down assays CutO also failed to associate with it (Supplementary Fig. 11b). It could be that the chromatography is a too aggressive method disassembling unencapsulated CutC–CutO complexes. Alternatively, it is also possible that CutC and BMC shell forms a composite binding site for CutO. This, however, would result in CutO being localized exclusively on the inner shell surface and thus being encapsulated in relatively small amounts, especially, when the larger volume/surface ratios of native-type BMCs are considered. Such mechanism would result in a very uneven distribution of CutO in BMC.

Of all three tested core enzymes, CutC was the only enzyme capable of co-migration with BDPs by itself when co-expressed

with cmcABC′ + D shell genes (Supplementary Fig. 13a). The encapsulation of CutC was further confirmed by His6x tag capture experiments. Both large and small type BDPs containing encapsulated His6x-tagged CutC could not bind to the HisTrap column, although free CutC could be efficiently purified by this method[45], suggesting that the CutC along with the His6x tag is sequestered inside the BDP lumen (Supplementary Fig. 19a, b). This was true for both small type and large type BDP materials. The presence of 6xHis tag in BDP-associated CutC was confirmed by western blot (Supplementary Fig. 19c). Neither CutO nor CutF was able to co-migrate with BDPs in detectable amounts. This was somewhat unexpected, considering that CutF (but not CutO) has an EP-like elongation (NEQNVERVIRQVLERLG) at the C-terminal end[56]. CutH contains an N-terminal EP-like elongation (MIDTLVREKIAARL)[56], but we failed to detect any CutH co-migration with BDPs as well. The presence or absence of cmcE had no influence on co-migration—CutC was present, and CutO and CutF absent irrespective of the presence or absence of cmcE (Table 2, Supplementary Fig. 13d–f).

An unexpected observation was that both CutO and CutF co-migration with BDPs could be observed if they were co-expressed with CutC (Table 2, Supplementary Fig. 14). Testing the N- and C-terminal domains of CutC revealed that the N-terminal domain was not essential for the encapsulation of CutC C-terminal part and the encapsulation of the N-terminal domain as such could not be detected either (Supplementary Fig. 15a, b). Thus, the first 335 amino acids of CutC are not necessary for its encapsulation. The presence of the CutC N-terminal domain was also not needed for the encapsulation of CutO, which could be encapsulated by both full-length CutC and truncated CutC$_{336-1128}$. However, the N-terminal domain was essential for the CutC-mediated co-migration of CutF with BDPs, which further highlighted the role of the N-terminal CutC domain (Supplementary Figs. 14 and 15). The presence of CutF changed the co-migration pattern; while encapsulated CutC and CutC + CutO were evenly spread throughout all BDP-containing fractions after gel filtration, indicating that there was no preference for a certain particle size, CutF co-migrated predominantly with large type BDPs (Supplementary Fig. 15b, c). Remarkably, the presence of CutF even shifted a portion of the CutC and CutO proteins from the small type BDP zone into large type BDP particle zone (Supplementary Fig. 15c). When the CutC + CutF + CutO proteins were co-expressed with the cmcC′ + D and cmcC + D constructs, capable of forming only small type BDPs, no co-migration of any protein was detected, suggesting that the core size could be too large for encapsulation in small type particles. However, CutC alone could be encapsulated in cmcC + D particles very efficiently (Supplementary Fig. 16). To ensure proper identification, the identity of the encapsulated CutO and CutF bands were additionally confirmed by peptide mass fingerprinting analysis (Supplementary Fig. 20).

Our observations of CutF migration in co-expression with BDPs are consistent with the observations for purified core enzymes (Supplementary Fig. 17a)—in both cases CutC and CutF interaction had a size-increasing effect. We also concede that some proportion of CutC + CutF and CutC + CutF + CutO could be present in a free form intermixed with BDPs in the case of coexpressions with BDPs, since the presence of CutC could be observed in small amounts mixed with BDPs as well (Supplementary Fig. 12e). Thus, encapsulation of the core enzymes in the presence of CutF is not certain.

Nevertheless, our data strongly suggest that CutC acts as a mediator of the encapsulation of the enzymatic core. It must be noted that our recombinant BDP system may not completely accurately represent native encapsulation system of *K. pneumoniae*, but it is very likely that CutC plays the central encapsulation

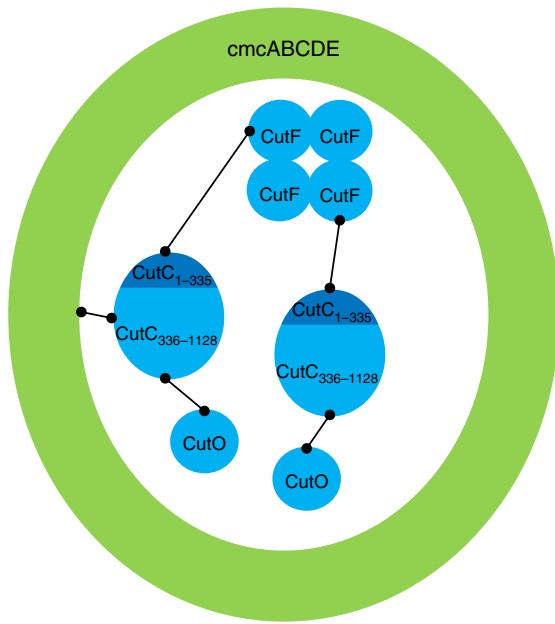

**Fig. 3 Proposed enzymatic core encapsulation mechanism of CutC, CutF, and CutO in GRM2 BDPs.** CutC is serving as an adaptor for the encapsulation of other enzymes. CutC C-terminal domain is responsible for encapsulation and also for interaction with CutO. The CutC N-terminal domain is responsible for CutC interaction with CutF. CutF together with CutC N-terminal domain crosslinks the enzymatic core and increases its size.

role in native conditions as well and the interaction between the CutC N-terminal portion and CutF is most likely necessary for the assembly of the native, large-sized enzymatic cores. An analog of CutF from a different BMC locus has been demonstrated to form a tetramer in a crystal structure[35]; therefore, it could serve as a multivalent cross-linker in the BMC core. Our proposed enzymatic core encapsulation mechanism is summarized in Fig. 3. The C-terminal part of CutC anchors itself and the entire enzymatic core to the shell and simultaneously ensures the encapsulation of CutO, while the N-terminal part of CutC ensures the encapsulation of CutF, which then further crosslinks the enzymatic core and increases its size.

**Cryo-EM characterization of BDPs**. We analyzed cmcABC′D + CutC$_{336-1128}$ BDPs with cryo-EM. The peak containing the small type BDPs in the gel filtration volumes from 90 to 100 ml was chosen for analysis because this material appeared to be the most uniform in TEM analysis (Fig. 2) and contained encapsulated CutC$_{336-1128}$ (Supplementary Fig. 6B).

We calculated a near-atomic (3.3 Å) resolution map for the icosahedral particles and built an atomic model (Fig. 4). The particles had a pT = 4 quasi-symmetry with 12 cmcD pentamers occupying the vertices of the icosahedron and 180 BMC-H monomers that were arranged in 30 hexamers within the facets. The cmcABC′ + D small type particle peak contains all three BMC-H proteins; most likely, they all contribute to the averaged electron density of BMC-H positions. The map is not of sufficiently high resolution to distinguish between these three types of proteins, so we chose to model cmcC′ in the model, since the mass spectrometry data identified it as the most abundant BMC-H protein in cmcABC′ + D small type particles (Supplementary Fig. 10a). Electron density for cmcC′ chain in pentameric–hexameric contacts was interpretable from residues 3 to 88, for the rest of cmcC′ chains in hexameric–hexameric contacts it was interpretable from residues 3 to 83. The resulting

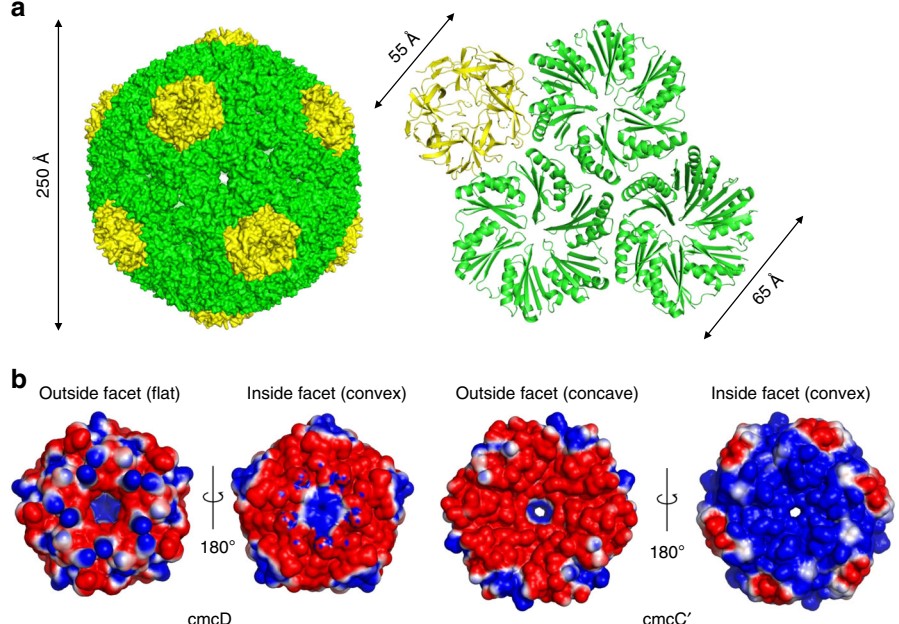

**Fig. 4 Cryo-EM structure of pT=4 quasi-icosahedral BDP and its pentameric and hexameric components.** **a** Surface model of pT = 4 quasi-icosahedral BDP particle, displayed on the left side. A ribbon model of a cmcD pentamer and three cmcC′ hexamers is displayed on the right side. Pentameric cmcD protein is colored in yellow and hexameric cmcC′ is colored in green. Note that the fivefold symmetry axis is located at the center of cmcD pentamer and threefold axis is located in the middle between three cmcC′ hexamers. **b** Electrostatic surface potential of pentameric cmcD and hexameric cmcC′. Note the pores in the centers of pentamers and hexamers. The surface contour levels were set to $-1\ kT/e$ (red) and $+1\ kT/e$ (blue).

pT = 4 BDP particles are 250 Å in diameter. These particles are formed similarly to previously reported larger pT = 9 pseudo-symmetric particles from *Haliangium ochraceum*[28], pT = 4 particles comprised double-fused *Haliangium ochraceum* BMC-H proteins[57], mixed icosahedral and elongated pT = 4 and pT = 3 particles from *Halothece* sp[58], and smaller T = 1 particles comprised circularly permutated BMC-H proteins[59].

In the pT = 4 BDP shell, there are three pores: one is in the center of the cmcC′ hexamer, the second is at the threefold axis between the cmcC′ hexamers, and the third is in the center of the cmcD pentamer (Fig. 4). The pore in the center of the hexamer is approximately 7 Å in diameter as judged after subtraction of atomic Van der Waals diameters and contains typical GSG pore motifs on the rim (Fig. 4b, Supplementary Fig. 8). The pore in the center of the cmcD pentamer has a funnel shape and is approximately 16 Å wide on the outer surface but is only approximately 4 Å in diameter on the inner surface as judged by the Van der Waals diameters of atoms forming it (Fig. 4b). In electrostatic surface potential map it is visible that the cmcC′ hexamer inner facet is charged more positively than the outside facet (Fig. 4b). The convex facets of cmcC′ and cmcD are directed towards the lumen, a feature observed in earlier studies of other types of BMCs[28,57,58]. The narrowest part of the pentameric pore is very hydrophobic in nature, as it is lined by tyrosines and phenylalanines (Supplementary Fig. 21). Such a ring-like structure of aromatic amino acids in the BMC-P pore is unusual and seems to be a characteristic of the GRM2-type (Supplementary Fig. 8). This feature is not completely conserved, as *Aeromonas hydrophila* GRM2-type BMC-Ps have only one aromatic amino acid instead of two in the matching central pore forming motif (Supplementary Fig. 8). Most BMC-P proteins encoded by other loci contain a small helix between the β5 and β6 strands, and the rim of the pore is lined by GS motifs to form a pore of comparable size. Beta carboxysomal shell protein CcmL can be illustrated as an example of this typical fold (Supplementary Fig. 21). Compared to BMC-P proteins from other hosts, the GRM2-type BMC-P sequence contains a deletion of five amino acids downstream from the pore motif, which thus transforms the small helix between the β5 and β6 strands into a shorter loop. In our structure the tip of this loop is disordered—residues K66, D67, and R68 and side chains of L64, N65, Y69, and K70 are invisible in electron density and are not modeled. The bulky aromatic amino acids seem to compensate for the shorter loop in terms of closing the pore. Remarkably, the GRM3-type BMC-P contains the same five amino acid deletion but retains the typical GS pore motif, so the GRM3 BMC-P pore would be significantly wider than that of any other BMC-P (Supplementary Fig. 8). The third pore at the BDP threefold axis is 3 Å in diameter and mostly featureless, as it is formed by the nearby main chain of glycine residues. Substrate and cofactor transport across the shell is probably mediated by the largest cmcC′ pore since the other two types of pores are too small or hydrophobic to be involved in this process, if it is assumed that the pores are mostly static in their conformations.

The cryo-EM analysis revealed significant heterogeneity among the particles (Fig. 5). While the majority of particles were revealed to be pT = 4 pseudosymmetric icosahedrons, roughly a quarter of these pT = 4 particles were missing at least one pentameric unit, resulting in an incomplete particle. There were also smaller quantities of elongated pT = 4, Q = 6 and pT = 4, Q = 8 particles, and there were more elongated filamentous particles and icosahedrons with larger pseudosymmetry numbers (possibly pT = 7 or pT = 9). A very unusual subpopulation consisted of smaller triangular-shaped particles, which, judging by their size, could be derivatives of three fused pT = 4 icosahedrons (Fig. 5d). Several asymmetric C1 classes of icosahedral particles with visible electron density inside the BDP particle could also be identified (Supplementary Fig. 22), which we hoped contained encapsulated CutC_{336-1128}. However, despite extensive efforts, including particle subtraction, we failed to generate any meaningful reconstructions beyond 20–30 Å resolution of the object inside the particles. There are several possible reasons for this result.

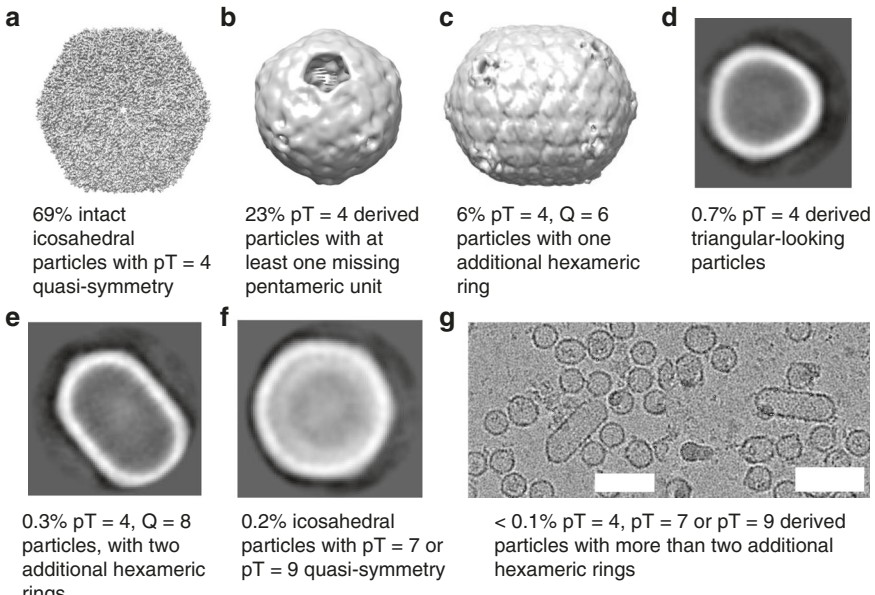

**a** 69% intact icosahedral particles with pT = 4 quasi-symmetry

**b** 23% pT = 4 derived particles with at least one missing pentameric unit

**c** 6% pT = 4, Q = 6 particles with one additional hexameric ring

**d** 0.7% pT = 4 derived triangular-looking particles

**e** 0.3% pT = 4, Q = 8 particles, with two additional hexameric rings

**f** 0.2% icosahedral particles with pT = 7 or pT = 9 quasi-symmetry

**g** < 0.1% pT = 4, pT = 7 or pT = 9 derived particles with more than two additional hexameric rings

**Fig. 5 Cryo-EM classes of BDP subtypes. a** 3D class at 3.3 Å resolution and atomic model of whole intact pT = 4 quasi-symmetric icosahedral particles, in total 69%. **b** 8.8 Å resolution 3D class of pT = 4 derived particles, with at least one missing pentameric unit, in total 23%. **c** 9.6 Å resolution 3D class of pT = 4, Q = 6 quasi-symmetric particles, in total 6%. **d** 2D class of triangular-looking particles, judging from the size probably derived from three fused pT = 4 particles, in total 0.7%. **e** 2D class of pT = 4, Q = 8 particles, in total 0.3%. **f** 2D class of pT = 7 or pT = 9 quasi-icosahedral particles, in total 0.2%. **g** Separate micrographs of tubular particles elongated with more than two hexameric rings, in total less than 0.1%. Scale bar: 60 nm.

First, the number of such particles is quite small, with only 3–5 thousand particles in each class. Additionally, choline-free CutC is partially disordered, as demonstrated in our previous research[46], and this could complicate the process of the sorting of proper two-dimensional (2D) classes. In addition, it is also possible that encapsulated CutC is not rigidly bound at a particular position within the BDP shell. If this is indeed the case, the CutC protein acts as a nucleator of shell assembly. Still, the fact that the unidentified electron density was located in the inside of the particles suggests that some kind of cargo is encapsulated, and it can reasonably be suspected as CutC.

The minor BDP shell subpopulations illustrate the flexibility of BMC-H proteins in creating different shapes, including the ability to form curved surfaces in small tubular particles, more planar surfaces in larger icosahedral particles and semi-regular poly-hedrons by fusing separate icosahedrons into more complex formations. This is not surprising since native metabolosomes, when visualized in TEM, have large, irregular shapes[12,13,15]. The multiple building modes are a result of the flexible contacts between BMC-H hexamers that range from 150° in pT = 4 icosahedrons to 160° in the elongated pT = 4, Q = 6 particles (Supplementary Fig. 23) to a planar 180° (ref. [21]), as has been previously demonstrated in the crystal structure of a BMC-H protein homolog.

The main hexamer–hexamer contacts between cmcC′ are formed by the conservative K–R–X triad (Fig. 6a, Supplementary Fig. 24), which has already been demonstrated in previous reports[21,22]. A hydrogen bond is formed between the K25 side chain nitrogen and the main chain of R78 in the adjacent monomer. The R78 side chain can in turn form a hydrogen bond with the main chains of K25 and A27 in the opposite monomer. These interactions result in a ring structure of four linked cmcC′ monomers. On the inside surface, there is also a probable network of salt bridges formed by the E62, R66, E16, and R74 residues. The side chain electron density is very poor for this network, and it is impossible to visualize the exact contacts between the particular residues, but their positions in the main chain suggest

that interactions between them do occur. These residues are not completely conserved, but most cmcC BMC-H homologs contain several basic and acidic residues in the same regions (Supplementary Fig. 8). This salt bridge network could play a significant role in controlling the planarity of the contacts between the hexameric units; for example, the distances between the amino acids at the E62 and R66 positions in opposite monomers can vary significantly from 8 Å in a planar crystal contact, as shown previously[21], to almost 3 Å in the 150° contact found in our pT = 4 BDP structure.

There are also several specific polar pentameric–hexameric cmcC′–cmcD interactions (Fig. 6b). There are hydrogen bonds between the cmcD G49 main chain and the cmcC′ K25 side chain and between cmcD E79 side chain and cmcC′ G51 side chain. There are also two possible salt bridges between the cmcC′ R78 and cmcD D54 and the cmcC′ D49 and cmcD K80 side chains. The solvent-accessible buried surface areas between the hexameric–hexameric and the pentameric–hexameric contacts are very similar (1050 and 1010 Å², respectively). Nevertheless, in our case, roughly a quarter of the pT = 4 BDPs were missing at least one pentameric subunit, a somewhat surprising fact when the numerous pentameric–hexameric contacts and comparable buried surface areas are considered. In another case, a pT = 4 BMC shell particle was formed by a double-fused BMC-H protein even in the absence of any BMC-P proteins[57]. This demonstrates that BMC-P proteins are much more loosely integrated into GRM2 BDPs than BMC-H proteins and has potential implications for designing synthetic BMC particles with encapsulated cargo.

## Discussion

Our results demonstrate that even highly similar BMC-H proteins may have significantly different assembly properties. A computational study has shown that shell proteins with high spontaneous curvature are more efficient in forming empty particles[60]. High spontaneous curvature components would form nanotube-like or

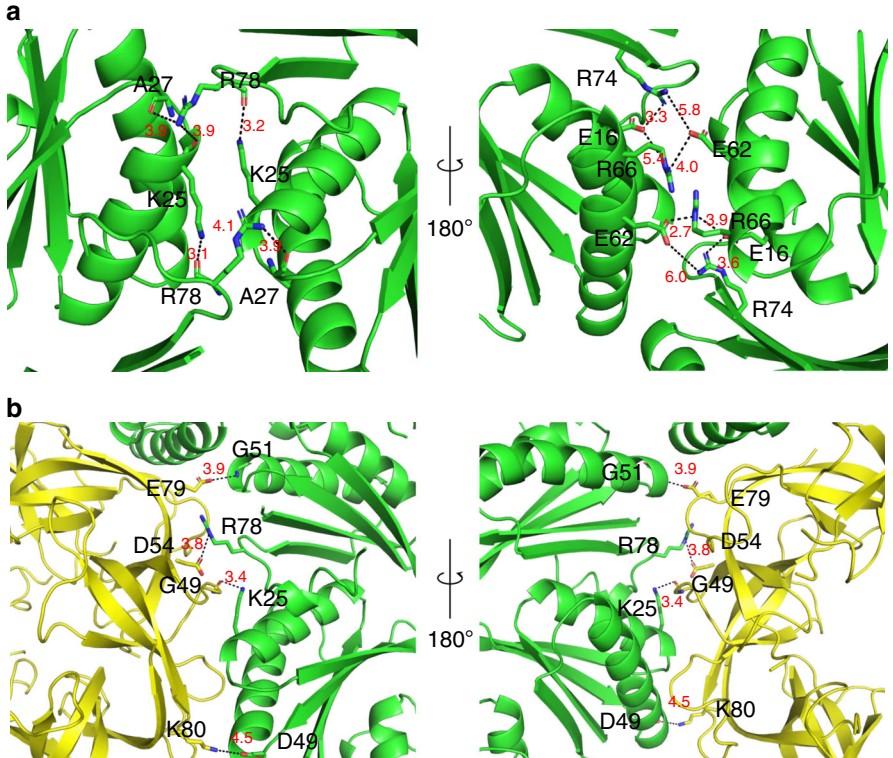

**Fig. 6 Detailed view of hexameric–hexameric and pentameric–hexameric interfaces. a** Hexameric–hexameric cmcC′–cmcC′ interface; K–R–X (in our case K–R–A) triad and salt bridge networks are viewed as stick models, distances are measured in angstroms. **b** Hexameric–pentameric cmcC′–cmcD interface; amino acids involved in contacts are viewed as stick models. Distances are measured in angstroms.

irregular assemblies in the absence of pentameric subunits or small icosahedral particles in the presence of pentameric subunits, and the likelihood of purifying such material with our method would be increased. Low spontaneous curvature components would form large planar surfaces that would be too insoluble or unstable to be purified with our methods. If our experimental results are considered in this light, then individual cmcA and cmcB would correspond to components with low spontaneous curvature, and cmcC′, cmcE, cmcC_trunc and, to a lesser extent, native cmcC would correspond to high spontaneous curvature components. It is possible that some synergic effects could also be involved in the determination of curvature—cmcAB + D are able to form small type particles in contrast to individual cmcA + D and cmcB + D unable to do so. If this hypothesis is indeed the explanation of our results, the formation of larger and more native-type BDP shells requires a fine balance between separate high-curvature and low-curvature components.

Recombinant BMC shell particles are sometimes smaller and more regular than native BMCs in their natural hosts[28,38,39,53,61], as was also demonstrated in our case. In our recombinant expression system, larger, more complete enzymatic core (CutC + CutF + CutO) does not enlarge the shells of predominantly smaller BDPs (cmcC + D) and, as a result, are not encapsulated. This observation may not be true for formation of native BMCs where the expression and assembly processes could be significantly different and the complete native core itself could be a significant factor in particle assembly and size determination.

It is rather unclear exactly which part of the protein or which set of interactions could be playing the role of the "switch" between the high-curvature and the low-curvature BMC-H components in our case. The nature of these components is not necessarily individual—for example, formation of particular

heterohexamers of different types of BMC-H proteins could be possible. A good candidate for the "switch" role would be the salt bridge network located on the lumenal interface between the hexamers since it shows variability among cmcC, cmcA, and cmcB and between different BMC types, yet it can still be identified in most of the short BMC-H proteins (Supplementary Fig. 20). Another potential candidate for this role could be the BMC-H C-terminal portion beyond residue 85. C-terminally elongated BMC-H proteins (cmcC′ and cmcE), at least in our case, seem to be crucial in forming larger assemblies; however, in the case of cmcABC_trunc + D a C-terminal truncation also had a similar enlarging effect on the size distribution of BDPs. Since the C-terminus is located close to the interhexamer contacts, it is tempting to speculate that it might also somehow control the contact angle between the hexamers in such a way that it regulates the size of the final particle. Remarkably, there are reports that describe another case of an artificially created C-terminally elongated BMC-H mutant, designated PduA*[62,63]. In this case, the solubility of the mutant PduA* was greatly enhanced, and it was also more capable of nanotube-like structure formation in E. coli. Although C-terminal elongation does not seem to improve the solubility of GRM2 BMC-H proteins, it is indeed associated with formation of elongated nanotubes for cmcE and cmcABC′ and also with formation of elongated BDPs. Depending on its binding partner, the cmcC′ C-terminal helix can assume different conformations; in a hexamer–pentamer contact, it is more structured than in a hexamer–hexamer contact (Supplementary Fig. 25). A somewhat analogous mechanism exists in some icosahedral plant and animal viruses, where an N-terminal "arm" containing coat protein is assembled and inserted between monomers in hexameric contacts, which makes them flat but results in disordered pentameric contacts that are bent, which was

first noticed in tomato bushy stunt virus (TBSV)[64]. Whether this actually contributes to the contact angles between BMC-H hexamers remains unclear.

The assembly of the enzymatic core is a crucial step in BMC formation. Carboxysomes contain specific adaptor proteins, the purpose of which is to ensure the crosslinking and encapsulation of the enzymatic core[33,42–44]. This is not the case for metabolosomes, where there are no specific additional adaptor proteins and encapsulation must be carried out by EP that is directly attached to enzymes. However, not all core enzymes contain identifiable EP, and it has been hypothesized previously that a piggybacking mechanism ensures the encapsulation of the core components without EP[56]. Experimental observations supports this as well—activating enzyme of GRM3 signature enzyme binds to the signature enzyme and is probably encapsulated in such manner[36]. There is also some evidence of EP-independent interaction of core enzymes with the BMC shell—for example, PduS interacts with one of the *Citrobacter freundii* pdu shell component PduT without mediation of an identifiable EP[65]. There is a benefit to a piggybacking strategy, as it would provide an opportunity to control the stoichiometry of encapsulated enzymes and ensure that no component is present in such low quantities as to bottleneck the enzymatic cascade.

Our experimental results show that this mechanism could be true for GRM2 enzymatic core components as well, as the interactions between some GRM2 core components are strictly hierarchical. Encapsulation is dependent on sequential interactions between particular enzymes, where CutC is responsible for core encapsulation in the shell and, together with CutF, is most likely responsible for crosslinking the core into a larger assembly (Fig. 3). It is interesting that $CutC_{336-1128}$ has no identifiable EP sequence and therefore results in encapsulation that seems to be different from that resulting from the canonic EP–shell interaction. A reason for this could be the lack of BMC-T proteins in the GRM2 locus. BMC-T PduB has been shown to be an essential component for EP-mediated encapsulation of cargo[40]; because of this, an entirely different BMC encapsulation system may have evolved in the GRM2 locus to compensate for the absence of BMC-T. Interestingly, CutF still contains an EP-like sequence at the C-terminal end and CutH has an analogous sequence at the N-terminal end, but their functions in our recombinant BDP system, if any, are limited purely to crosslinking. We failed to detect any CutH encapsulation or co-migration with BDPs or core enzymes, although it is vital for regeneration of coenzyme A inside the lumen and most likely is encapsulated in native BMCs[66]. The observations are not necessarily true for native GRM2-type BMCs—it is possible that EPs of CutF and CutH become functional only in native conditions. Nevertheless, the role of the CutC as a central adaptor stays strongly suspected in native GRM2 BMCs as well. The encapsulation pattern could point to another important consideration for metabolosome organization. An independent encapsulation of CutO and CutF mediated by CutC could be a good way to ensure that both alcohol and aldehyde dehydrogenase are in close proximity to the signature enzyme. Proper $NAD^+/NADH$ recycling inside the enclosed compartment is based on the requirement that the aldehyde intermediate is more or less equally divided between the alcohol and aldehyde dehydrogenase components[9,20], and this could be a good mechanism for ensuring this.

## Methods

**Construct design.** GenBank™ entry ARRZ01000032.1 was used for PCR primer design, with the following entries that correspond to named proteins: CutC (EPO20241.1), CutO (EPO20327.1), CutF (EPO20363.1), cmcA (EPO20272.1), cmcB (EPO20357.1), cmcC (EPO20328.1), cmcE (EPO20271.1), and cmcD (EPO20293.1). cmcB had to be obtained via separate gene synthesis by General

Biosystems (USA), since it was not possible to amplify it due to interference from the cmcA and cmcC genes. Two additional constructs were made from full-length CutC: $CutC_{1-325}$, corresponding to the first 325 amino acids of CutC, and $CutC_{336-1128}$, corresponding to amino acids 336–1128. Constructs containing cmcABC and cmcABC′ were generated by amplifying and cloning the whole cmcABC region with forward cmcA and reverse cmcC primers. Genomic DNA obtained from a *Klebsiella pneumoniae* strain MSCL535 (Microbial Strain Collection of Latvia) was used as a PCR template. The primer pairs used are listed in Supplementary Table 1. To ensure maximal simplicity in the design and co-expression, the proteins were expressed in a pET-Duet1/pRSF-Duet1 two-plasmid system. These plasmid vectors contain identical Duet regions that contain dual promoters and can also be cotransformed. DNA transcribed under the T7-1 promoter was inserted using *Nco*I and *Hind*III sites, and for insertion after the T7-2 promoter, *Nde*I and *Xho*I sites were used. To increase the number of available promoters from four to six for cmcABC′ + D + CutC + CutF + CutO and cmcABC′ + D + CutC + CutF + CutO + CutH constructs, we amplified the whole Duet region containing CutO or CutO + CutH and inserted it at the end of the pRSF-Duet1 vector at the *Xho*I site.

**Protein expression and purification.** pET-Duet1 plasmids containing cmcABC + cmcD and, optionally, pRSF-Duet1 plasmids containing CutC/CutO/CutF/cmcE were transformed into *Escherichia coli* BL21-DE3 chemically competent cells (Sigma-Aldrich, cat. No CMC0014). Cells were grown in 2xTY medium containing 50 μg/ml ampicillin and (if pRSF-Duet1 vector was used) 30 μg/ml kanamycin. Cells were grown at +37 °C to OD₅₉₀ 0.7 and shaken at 200 r.p.m., cooled at 20 °C for 30 min and induced with 1 mM IPTG. Induction was performed overnight for approximately 16 h at 20 °C with shaking at 200 rpm. The biomass was then collected, centrifuged, and frozen at −20 °C.

Cell lysis was performed in a buffer containing 100 mM Tris-HCl (pH 8.0), 300 mM NaCl, 0.1% Triton X-100, 20 mm MgSO₄, 0.1 mg/ml DNase, 1 mg/ml lysozyme, 1 mM PMSF, and 2 mM DTT. The lysate was incubated at +6 °C with shaking for 1 h and then centrifuged at 10,000g for 10 min. The supernatant was then collected and centrifuged at 50,000g for 3 h. The supernatant was discarded, and the pellet was suspended in a small volume of 300 mM NaCl and 20 mM Tris-HCl (pH 8.0), maintaining a proportion of 4 ml per 5–10 g of the initial cell biomass. The suspension was then centrifuged at 10,000g for 10 min, and the supernatant was collected. The solution was then loaded on a 16/900 Superose 6 gel filtration column (GE Healthcare) equilibrated in 300 mM NaCl and 20 mM Tris-HCl (pH 8.0), and 2 ml fractions were collected. Fractions 30–60 (corresponding to 60–120 ml), which contained the BMC proteins, were analyzed with 15% SDS-PAGE and TEM.

His6x-CutC, His6x-CutC₃₃₆–₁₁₂₈, His6x-CutH, His6x-CutF, and His6x-CutH purification and BDP capture tests were performed by Ni²⁺ affinity chromatography. The frozen biomass was suspended in lysis buffer containing 100 mM Tris-HCl (pH 8.0), 200 mM NaCl, 1% Triton X-100, 1 mM PMSF, and 2 mM DTT. Cells were lysed by ultrasound, and the lysate was centrifuged at 14,000g for 40 min. CmcE and CutO were purified on a 1-ml HisTrap column (GE Healthcare). For this step, 20 mM imidazole in 40 mm Tris-HCl (pH 8.0) and 300 mm NaCl was used as a washing buffer, and 300 mm imidazole in 40 mm Tris-HCl (pH 8.0) and 300 mm NaCl were used for the elution buffer. The cmcABC′ + D + CutC and cmcABC′ + D + CutC + CutF + CutO capture tests were performed in the same manner; all fractions were equalized by volume and the volume of the sample loaded on the SDS-PAGE gel was equalized as well. Gel filtration experiments were repeated at least twice to confirm the observations.

Western blot analysis of His6x-CutO and BDP-encapsulated CutC were perfomed by using His•Tag Antibody HRP Conjugate Kit (Novagen, cat. No 71840-3). The samples were loaded on a 8% SDS-PAGE gel and transferred afterwards to a nitrocellulose membrane in a semi-dry fashion. The blot was visualized with ECL Prime chemiluminescent detection reagent kit (GE Healthcare, cat. No RPN2232).

**Mass spectrometry analysis.** The SDS-PAGE bands of CutO, CutF, and the individual components of BDPs were identified by the peptide mass fingerprinting method. The protein band was cut out from the Coomassie blue-stained poly-acrylamide gel and washed twice for 1 h with 500 μl of 0.2 M ammonium bicarbonate and 50% acetonitrile. Then, the gel fragments were washed twice with 200 μl of 100% acetonitrile and incubated with a trypsin (Sigma-Aldrich, cat. No T6567) solution containing 40 mM ammonium bicarbonate and 10% acetonitrile for 3 h at +37 °C. A total of 1 μl of the obtained peptide solution was mixed with 1 μl of 0.1% TFA and 1 μl of matrix solution containing 15 mg/ml 2,5-dihydroxyacetophenone in 20 mM ammonium citrate, and 75% ethanol. For BDP analyses, samples from particular fractions of gel filtration on Superose 6 were used directly instead of peptide solution. Then, 1 μl of the obtained mixture was loaded on the target plate, dried, and analyzed using a Bruker Daltonics Autoflex MALDI-TOF mass spectrometer.

**TEM analysis.** BDPs were visualized by TEM with uranyl acetate negative staining. A 5 μl sample drop was placed on a formvar-coated TEM 200 copper grid (Sigma-Aldrich) and incubated for 3 min. The grids were dried, briefly washed with 1 mM

EDTA solution, and negatively stained with 1% uranyl acetate for 1 min. The grids were then dried and analyzed on a JEM-1230 TEM electron microscope at 100 kV.

**Cryo-EM analysis of BDPs and model building.** A total of 4 µl of purified cmcABC′ + cmcD + CutC$_{336-1128}$ BDPs with a concentration of 1 mg/ml in 100 mM NaCl and 20 mM Tris-HCl (pH 8.0) were applied to the EM grid. The grids (Quantifoil, Cu grids, 200 mesh, R2/1) were blotted for 4 s using a Vitrobot (Mark IV, Thermo Fisher) at 18 °C and in 100% humidity, plunge-frozen in liquid ethane–propane and stored in liquid nitrogen until further use. Cryo-EM data were collected with a 200 kV Talos Arctica microscope (Thermo Scientific) equipped with a Falcon 3EC direct electron detector (Thermo Scientific). A total of 1316 images (Supplementary Fig. 26) was collected in an automated manner using EPU software (Thermo Scientific). The data were collected at the nominal magnification of ×120,000, corresponding to a calibrated pixel size of 1.23 Åpx$^{-1}$, with an underfocus in the range of −1.4 to −3.0 µm and an exposure time of 1.0 s, comprising an overall dose of 60 eÅ$^{-2}$ for each specimen. Data from a single exposure were stored as a set of 40 movie frames.

Motion-correction and dose-weighting of the frames were performed in MotionCor2 (ref. [67]), and CTF correction was performed with Gctf[68]. The single particle analysis was performed with the RELION 3.0 pipeline[69]. The general scheme used for the analysis is given in Supplementary Fig. 13. Approximately 900 particles were picked by log-based autopicking. After several rounds of reference-based autopicking and a final manual inspection, a total of 62,533 particles were selected. A subsequent 2D classification was performed that illuminated a noticeable heterogeneity in the BDP particle morphology. Five 2D classes with a total amount of 45,915 particles were selected for three-dimensional (3D) de novo model generation in I point group symmetry. The obtained icosahedral model was then used in two 3D classifications—in I and C1 point group symmetries. The icosahedral model was 3D refined and postprocessed, and the refined map reached 3.8 Å resolution. After further CTF refinement and Bayesian polishing cycles, the refined particles were used for another high-resolution 3D refinement, reaching a final resolution of 3.3 Å. The final map was sharpened by applying a b-factor of −149.892 Å$^2$. Two additional lower resolution maps were made; one was in C1 point group symmetry based on two 3D classes of C1 classification and the other was in D5 point group symmetry based on two 2D classes of initial 2D classification (Supplementary Fig. 13). The C1 reconstruction was obtained from 11,639 particles and the D5 reconstruction was obtained from 3153 particles. The final resolutions after high-resolution 3D refinement were 8.8 Å for the C1 map and 9.6 Å for the D5 map. For all final 3D refinements, the resolutions were estimated using the 0.143 cut-off criterion with gold-standard Fourier shell correlation between the two independently refined half-maps.

High similarity X-ray crystallographic structures were available for both cmcD (PDB ID 4N8X) and cmcC′ (PDB ID 4QIV), so homology models built with SWISS-MODEL[70] were used as the initial models. The initial fitting of the models in the map was performed manually by using UCSF-Chimera[71]. The models were then further built manually using Coot[72] and refined with REFMAC5 in the CCP-EM package[73] and PHENIX[74]. Residues K66, D67, and R68 and side chains of L64, N65, Y69, and K70 of cmcD are invisible in electron density and are not modeled. Images were generated in PyMol. The overall statistics of the maps and the atomic model are listed in Supplementary Table 2. cmcC′ chain in pentameric–hexameric contacts was interpretable from residues 3 to 88 (out of 100 total residues), for the rest of cmcC′ chains in hexameric–hexameric contacts it was interpretable from residues 3 to 83 (out of 100 total residues). cmcD chain was interpretable from residues 1 to 84 (out of 88 total residues); also, residues K66, D67, and R68 and side chains of L64, N65, Y69, and K70 are invisible in electron density and are not modeled.

**Reporting summary.** Further information on research design is available in the Nature Research Reporting Summary linked to this article.

## Data availability
The associated macromolecular structural data are available in the Protein Data Bank repository as 6QN1 entry and in the Electron Microscopy Data Bank repository as EMD-4595, EMD-4596, and EMD-4597 entries. All other relevant data are included in the paper and its Supplementary Information files.

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

## Acknowledgements
This study was supported by University of Latvia Foundation grant "Bacterial microcompartments as synthetic nanoreactors" funded by SIA MikroTik and the State Research Program BIOMEDICINE. CIISB research infrastructure project LM2015043 funded by MEYS CR is gratefully acknowledged for the financial support of the measurements at the CF Cryo-electron Microscopy and Tomography CEITEC MU. We thank Dr. Janis Rumnieks for technical IT support and Dr. Reinis Rutkis for valuable discussions.

## Author contributions
G.K. designed and produced constructs, expressed and purified proteins, solved the cryo-EM structure, and wrote the paper. E.-E.C. designed and produced constructs, and expressed and purified proteins. J.L. wrote the paper. J.J. prepared negative staining TEM samples and performed TEM analysis. A.F. prepared cryo-EM samples and collected cryo-EM data. K.T. coordinated the study and wrote the paper.

## Competing interests
The authors declare no competing interests.
