## [Peer Review File · Nature Communications]

Reviewers' comments:

Reviewer #1 (Remarks to the Author):

The authors present a structural and enzymatic study of a highly miniaturized version of a glycy radical type bacterial microcompartment. In addition to the important structural details about protein-protein interactions in the shell, the study provides new information about motifs and domains that appear to organize and sequester the interior enzymes, and about molecular transport limitations conferred by the shell. The work represents an important addition to the bacterial microcompartment field. I have several concerns that should be addressed prior to publication.

Major points:

- 1) The assembly under study is an artificial representation of a native microcompartment. Compared to a wild type glycy radical microcompartment, the authors need to do more (in the discussion and perhaps elsewhere) to explain to the reader how differences in size, composition, and regularity should be considered. How might these differences affect interpretations of assembly geometry, enzyme encapsulation, transport, etc.
- 2) The implications about transport being limited by the shell are important, but these are points that are most open to criticism and in need of more careful elaboration. Early structure-based analyses of the likely degree to which the shell slows small molecule transport date to Tsai et al. 2007 (PMID 17518518). That analysis and subsequent modeling by Mangan et al. 2016 (PMID 27551079) have suggested that the shell provides a diffusive barrier to small molecules that could be modest but probably not a 50-fold effect, for example. The authors of the current paper get to the possible issue of cofactors vs small molecule transport near the end in the discussion, but they need to discuss earlier what their experiments might really be testing the transport of. It seems that the particular experimental setup is probably not cofactor-balanced, and how this weighs into the analysis needs to be made clearer earlier.
- 3) As possibly relative newcomers the authors do a reasonable job overall with referencing, but there are some gaps that should be remedied.
 - The first part of the introduction notes the range of BMC types. Jorda et al. 2013 (PMID 23188745) provided a bioinformatic articulation of the major BMC types that are now recognized.
 - In the second paragraph of the Introduction, pentameric vertex proteins are introduced. That discovery is due to Tanaka et al. in Science 2008, which should be credited.
 - The discovery that the main BMC shell proteins are hexamers is due to Kerfeld et al. in Science 2005.
 - The formation of flat sheets within crystals was noted in Kerfeld 2005, and the first demonstration of sheet formation outside of the crystal state is due to Dryden et al. 2009 (PMID 19844993).
 - The authors here use the name BMC-P for the vertex proteins. That protein family was first given the name BMV in 2013 (see Wheatley et al. PMID 23456886) in order to make clear that it is a different protein family from the BMC family. The BMC-P naming unfortunately loses that distinction. I suppose authors have leeway to choose naming conventions, but as a matter of priority if not clarity, it would be helpful to readers to make the equivalence clear by noting that that family is also known as BMV.
 - The third paragraph introduces the subject of glycy radical microcompartments. Jorda et al 2013 is where that important and diverse family was described. This paper has been overlooked in some recent papers on the glycy radical microcompartments; that should not be perpetuated here.

Minor points:

- Given how vaguely defined the meaning of resolution is for EM, it probably makes more sense to

keep the resolution value limited to one digit after the decimal, i.e. 3.3A.

- Is glycerol favored over glycerine as a common chemical name?
- There seems to be an extra sentence fragment at line 207.
- On line 285, the authors describe the protein interactions as 'interlocking'. If they mean to stick to that description, exactly what does it mean?
- The authors discuss shell protein pore sizes (line 323 and elsewhere). They need to be more specific about what they are measuring as the pore size. Is this the distance between atom centers across the pore? Or have the van der Waals radii of the atoms been subtracted?
- Figure 7 needs to be plotted a different way. As it is, the main observation is obscured.
- The use of 6 significant digits for the B-factor (methods) is excessive, by multiple digits.

Reviewer #2 (Remarks to the Author):

This is potentially an important contribution to the study of the structure and function of bacterial microcompartments, which are coming to be recognized as a major factor in microbial metabolic diversity and because of their outstanding potential for use in bioengineering.

The authors focus on a previously uncharacterized type of organelle, the GRM2, which has significant implications for human health. The paper reports an impressive amount of data on both the delimiting shell and the internal organization and assembly of the enzymatic core of the GRM2 BMC (elucidation of the function of the different domains of the CutC enzyme in encapsulation is a significant contribution toward the understanding of GRM function, and BMC biogenesis in general), however some of the other conclusions should be reconsidered and tempered. If queries about aspects of the experimental work can be satisfactorily addressed and, overall, the findings could be better contextualized with the current understanding of BMCs, especially GRMs (a recent review by Ferlez et al (Mbio 2019) could provide a ready handle on this surprisingly abundant but poorly characterized family of BMCs that encapsulate glycyl radical enzymes, including the different structural features of the GRE (cutC here) that seem likely related to assembly, this contribution to the field is certainly worthy of publication in Nature Communications.

Shell Characterization and Structure: This is a solid contribution about what can be formed from expressing different contributions of shell proteins outside of their native context and is done here in a completely new system. The authors should consider the pioneering work of Parsons et al., in this and also previous observations of nanotubes from shell proteins—this also relates to their shell particle classes.

Regarding the structure of the synthetic shell (needs to be clarified—this cannot be considered a proxy for the shell of an intact GRM2 metabolosome, much less an intact metabolosome (e.g. line 96 "minimum requirement for BMC formation; this is inaccurate— "BMC" should be reserved for shell + cargo, aka the native organelle. What has been obtained with cmcC+D is a minimal shell, it cannot even be considered a proxy for an empty native shell as it is missing the other shell proteins). Indeed, throughout the structure description it should be clear that this is an empty synthetic shell—likewise, calling it a particle can give the mis-impression that this is an intact BMC, or even a proxy shell containing all components; referring to "shell particles", or making a definition statement earlier on will prevent misunderstanding.

Please provide electron density figures of the discussed features, e.g. Supp. Fig. 12/pores (also in Supp Fig. 12 the protein name of 2QW7 is CcmL, not cmcL). Indicate which residues are modelled / which are disordered/missing for the shell structure.

The model geometry should be better considering the high particle symmetry (high clashscore,

10% rotamer outliers, 2.2% Ramachandran outliers), maybe try refinement with phenix.refine. Please comment on the concave-convex properties of the shell protein, and the sidedness of the shell in the structure. This is an important comparison to a distinct type of BMC shell that has also been determined by cryo-EM (Greber, Sutter, Kerfeld 2019)

In light of the cryo-EM structure, the question remains as to whether or not cmcA and cmcB are present in any of the GRM2 shells formed; would you obtain the same results (sizes, distributions, yields, etc.) if only cmcC'DE and were used to assemble shells? Do either cmcA or cmcB form homo-oligomeric structures that also have broad elution profiles on gel filtration that could be coeluting with the shells? The authors discuss and model the cmcC' hexamer and cmcD pentamer in the cryo-EM structure but do not mention the cmcA and cmcB hexamers included in the expression vector used to generate the 25 nm shells analyzed. Is it correct that the authors used the cmcABC'D +CutC336-1128 construct? If so, are cmcA and cmcB not present in these shells? This and the model quality (missing sidechains) and that the model is of insufficient resolution to distinguish paralogs should be made clear.

Considering the significance of the inter-hexamer gaps---this needs to be reconsidered in the context of the quality of the structure and, moreover, the non-native complement of shell proteins. Given its incompleteness in terms of structural building blocks, this is not a valid conclusion for a native shell. Likewise, the same incompleteness could potentially account for the failure to encapsulate certain components. The Kerfeld group has suggested that the binding sites for EPs, for example, may be composed of sites formed by specific combinations/organizations of shell proteins.

cmcB is not labeled in any main or supplementary figures. What is its expected molecular weight? Was it detected in shell containing gel filtration fractions? It is important to identify which of the expressed shell proteins are detected in each shell population as this will impact interpretation of encapsulation (e.g. inability to encapsulate CutH, or CutO/F in the absence of CutC) as well as structural properties of the resulting BMCs (e.g. does cmcA/B content effect size of the BMC shell?). Gel filtration results of individual shell proteins would also be useful to not only identify the same species in the gel filtration gels when other components are present but also detect any larger assemblies of a single species that may be co-migrating at the same size as assembled shells and so cause misinterpretation of what exactly is in the shell. Likewise, it would be useful to show the gel filtration profile of CutC alone to compare against the different elution patterns seen when coexpressed with the shell proteins.

Table1: There may be a typo here; the first two rows include two instances of the cmcABC + cmcD construct with different shell diameter distributions for each: the first states "predominantly 25 nm" and the second states "200-25 nm particles". What is the difference between these two constructs, is one meant to be cmcABC'?

Line 120: Figure S1d does not show cmcABC + cmcD + cmcE (instead the gel is labeled as cmcABC + cmcD +CutC + cmcE)

Fig. 2 What molecular weight do the peaks correspond to (compared to size standards; these should be presented), what is the measured absorption wavelength (A280?); later peaks seem low molecular weight but shells are observed in TEM. Figure 2a: please label location of BMC shell proteins in SDS-PAGE gel.

Figure 2b: it would be valuable to provide histograms and average size calculations for the various size shells observed by TEM for each zone collected from the gel filtration column. This data could offer insight into assembly mechanisms.

Lines 106-107 and 128-129: The authors attribute the increase in yield of the cmcABC'D BMC shells, as compared to the cmcABCD shells, to the C-terminal extension on cmcC'; were the

relative expression levels of *cmcC* and *cmcC'* examined? Could the increase in BMC shell formation be due simply to an increased cellular titer of *cmcC'* relative to *cmcC* as a result of the C-terminal extension? It's feasible to think that changes in the intracellular level of *cmcC'* could effect the stoichiometry of individual shell components such that the assembly of shells is favored and correspondingly more shells can be isolated during purification. To test the effect of expression levels of *cmcC* variants on shell assembly, the authors could use an inducible promoter for the expression of *cmcC* and show that shell yield and size does not vary as a function of induction (expression) levels.

As a negative control, was the *cmcC*trunc variant used to assemble *cmcABC*truncD shells to compare yields with *cmcABCD* or *cmcABC'D* shells? The relative expression levels of *cmcC*, *cmcC'*, and *cmcC*trunc should also be compared. The conclusion of 127-129 seems to be a bit of an overstatement.

A general comment about the shell composition of GRM2 (line 69); this is a fairly average number of shell protein paralogs for a BMC

Also, line 215-222: finding Hsp70 in an affinity purification of an over-expressed protein could also interpreted as adding to fold the over-expressed protein correctly. It is observed in overexpression of many distinct types of proteins (beyond BMCs). This observation is not particular to BMCs and this discussion should be removed.

Enzymology: The conclusions of the activity assay results are very speculative, it might be better to just show a decrease in the activity when FeADH is encapsulated by itself encapsulated but this could be due to many factors, e.g. NADH/NAD⁺ permeability, improper assembly of the BMC cargo due to the lack of the other proteins and potential over-encapsulation of CutC which could act as an additional diffusion barrier. Isolation and investigation of one step in a pathway that should operate as a whole can be misleading. This section in general needs to be tempered in its conclusions.

Fig. 7 plotting specific activity against enzyme concentration reveals little about the potential interesting effects going on; maybe consider experiments with variable Ethanol concentration

Line 154: Although the authors cite previous results where a his tag was successfully used to isolate CutC, did the authors confirm via western blot the presence of the his-tag on CutC in the shell-containing gel filtration fractions to confirm it wasn't degraded during the shell isolation protocol?

Lines 310-312: Were the 200 nm particles with CutC + CutF + CutO also assayed for CutO activity? Is the permeability of the larger shell any different than the smaller 25 nm shells shown? This would be interesting given the evidence that these small shells apparently contain only one hexamer (i.e. *cmcC'*) based on the cryo-EM data.

The alcohol dehydrogenase characterization as presented seems to simply show a relationship between concentration and activity. Since this cofactor may be recycled within the lumen in the native BMC, did the authors investigate the permeability of choline using CutC activity as a readout?

Assembly: Genetic fusions of such sequences have been demonstrated to be sufficient for the encapsulation of nonnative proteins during propanediol utilization (Pdu)^{31-32, 34} and ethanolamine utilization (Eut)³⁵ in beta-carboxysome³⁶ BMC particles." Based on the references cited, I believe the authors are citing the evidence for EPs to tag nonnative cargo in these three types of shells (Pdu, Eut, and carboxysomes) separately; as written, it sounds like proteins from Pdu or Eut systems have been targeted to the carboxysome which I don't believe to be true.

Line 145: Authors state no encapsulation of the PTAC (CutH) was observed; were cmcE containing shells used as well as the cmcABC(D) shells reported? Could the authors comment on the EP found on the PTAC (see Erbilgin et al., Plos Biology for background on this). This also relates to the surprising observation that the Aldehyde dehydrogenase was not encapsulated, despite containing an EP. The authors should comment on this. Also, regarding the solubility of the activating enzyme; perhaps it needs to be co-expressed with the signature enzyme, as shown previously for a GRM3 BMC (Zarzycki et al. 2017). This is also offers an example of "piggy-backing" as the authors observe in their GRM2 system.

It is a bit of over-interpretation of these data to state "the size of the 25 nm particles is not effected by the presence or absence of cargo and is likely limited by the shell composition"—or that the core is predetermined by the shell size. Lines 184-206: authors conclude that final BMC size is "determined by the intrinsic properties of the shell proteins and not by the core" (lines 194-196); however, the inability of the core enzymes CutC, CutF, and CutO to effect the size of cmcC+D particles may be due to structural limitations of this pair of shell proteins to form a polyhedral compartment larger than 25 nm. Is the size distribution of cmcABC'D shell particles effected by the presence of the enzymatic core enzymes (CutC, CutF, CutO)? Perhaps with the extra shell components (i.e. cmcA and cmcB) the presence of the core enzymes can influence the size of the resulting shells. If the size distribution is unchanged this would support the authors' claim that the size of the particle determines the packaging of the core, rather than being limited by the shell components. available.

Other:

- please provide explanations of the used abbreviations, e.g. Cut, cmc—coordinating colors and symbols across all figures would be helpful to the reader, especially the non-BMC community.
- please indicate molecular weights of the individual shell components and label corresponding bands on all SDS-PAGE gels
- Fig. 1a color pentamer differently, and use consistent coloring throughout
- Fig. 4b consider recoloring, maybe use electrostatics representation to provide more information
- line 233 the particles of the circularly permuted HO BMC-H are pT=3 (not pT=1)

Reviewer #3 (Remarks to the Author):

This manuscript describes the characterization of several aspects of a GRM-type BMC. The most impactful aspect of the manuscript is the structural data on the particles that formed from various expression experiments. Figures 4 and 5 are all of high relevance to the field right now. Findings such as the structures of minor BMC structures (eg elongated) and the high resolution of the structure make it a valuable contribution to the field. Unfortunately, other aspects of the manuscript are lacking, as detailed below. In particular, data and controls are lacking for most of the other experiments, or conclusions are drawn from data that is not included in the figures, and strongly worded claims are not supported with references to the data.

General/Main Text:

- 1) Many conclusions are made about the null outcome of an experiment in the manuscript. For example, in Line 162: "CutO and CutF were not encapsulated..." The data does not show that CutO and CutF are not encapsulated, but rather that they were not detected. All such mis-statements should be corrected.
- 2) Similarly, language needs to be softened for claims that are not directly supported by the

evidence. For example, Line 191-193, "It also seems that..." could be "One explanation for this result is that..." Or, "...the data suggest that the shell capacity..."

3) No protein interaction studies are used to confirm hypotheses about the role of CutC. Such support is required to support the putative explanations; gel filtration assays are not sufficient. While the experiments described are important first steps, this section by itself is not well-developed enough for inclusion as a main finding without additional supporting evidence.

Figure 1:

1) The authors state that cmcA, B, and C are "highly similar"; a multiple sequence alignment, either in this figure or in the supplement would be useful for identifying potential critical residues for compartment assembly.

2) Line 110: "empty volume" is usually referred to as "void volume"

Figure 2:

1) It would be useful if the authors would label the protein bands on the PAGE gel in 2A. Labels are present in the supplemental figures but not in the main text figure. The PAGE figures in the supplement should also have labelled molecular weight makers.

2) Authors claim that particles of particular sizes elute in certain fractions but no sizing data is presented. They should include quantitative data, for example by either measuring the diameter of imaged particles or doing DLS on these purified fractions.

Supplementary Figure 4d:

1) The authors claim that co-migration of CutC with BMC protein indicates encapsulation, but co-migration in SEC does not distinguish between luminal encapsulation, indirect association with the shell (via other co-eluting proteins, of which there are at least 20), direct binding to the shell exterior, or if it merely forms aggregates which are the same approximate molecular weight of the empty compartments (which vary dramatically from 25-200 nm in diameter). Any of these interactions when co-expressed could also shield effective purification using nickel affinity. The authors also do not show the HisTrap column elution profiles, which are central to their claims of encapsulation. The authors could effectively show encapsulation by fusing CutC to a fluorescent protein such as GFP and looking for fluorescent puncta in cells.

Supplementary Figure 5:

1) As with Supp Fig 4d, co-migration by SEC does not directly imply encapsulation, especially since the BMC shell proteins elute over such a broad range. Others have shown that BMC core enzymes natively aggregate. It's possible these enzymes aggregate into inclusion bodies that are in a size regime with a similar hydrodynamic radius to that of the empty BMCs. While these SEC results are promising, they are far from conclusive.

2) There are at least 20 other bands greater than 25 kD in MW, and the core enzymes make up a relatively small portion of the higher molecular weight bands present in the SEC profiles. This further supports their presence in nonspecific aggregates rather than inside of compartments.

3) The authors claim that the presence of CutF shifts encapsulation of the other enzymes towards the larger 200 nm compartments, but again, sizing data is absent. It is completely reasonable that inclusion of this protein increases aggregate size, rather than changing enzyme encapsulation preference for larger compartments.

Supplementary Figure 10:

1) Remove or even further soften claims. It is pure speculation that co-purification of an E. coli chaperone with a heterologously over-expressed unstructured BMC protein from K. pneumoniae implies that chaperones are necessary for compartment formation. It is much more likely that overexpressing an unstructured protein recruits chaperones.

Supplementary Figure 13:

1) The authors could not find any electron density indicative of encapsulated enzymes, perhaps because the enzymes are not actually encapsulated. This possibility is not considered, yet seems

the most obvious.

Figure 7:

- 1) The authors do not indicate whether there is a significant difference in activity between encapsulated and unencapsulated protein. There are also more points for unencapsulated than encapsulated enzyme, which is not explained.
- 2) There is no empty BMC control. There is also a surprising amount of activity at low enzyme concentration, possibly indicative of some baseline contamination.

Reviewer #1 (Remarks to the Author):

The authors present a structural and enzymatic study of a highly miniaturized version of a glycy radical type bacterial microcompartment. In addition to the important structural details about protein-protein interactions in the shell, the study provides new information about motifs and domains that appear to organize and sequester the interior enzymes, and about molecular transport limitations conferred by the shell. The work represents an important addition to the bacterial microcompartment field. I have several concerns that should be addressed prior to publication.

Major points:

1) The assembly under study is an artificial representation of a native microcompartment. Compared to a wild type glycy radical microcompartment, the authors need to do more (in the discussion and perhaps elsewhere) to explain to the reader how differences in size, composition, and regularity should be considered. How might these differences affect interpretations of assembly geometry, enzyme encapsulation, transport, etc.

We now address the implications of our results in context of difference between native and recombinant shells in results (lines 110-113) and discussion (lines 490-496, lines 547-549 and lines 564-568).

2) The implications about transport being limited by the shell are important, but these are points that are most open to criticism and in need of more careful elaboration. Early structure-based analyses of the likely degree to which the shell slows small molecule transport date to Tsai et al. 2007 (PMID 17518518). That analysis and subsequent modeling by Mangan et al. 2016 (PMID 27551079) have suggested that the shell provides a diffusive barrier to small molecules that could be modest but probably not a 50-fold effect, for example. The authors of the current paper get to the possible issue of cofactors vs small molecule transport near the end in the discussion, but they need to discuss earlier what their experiments might really be testing the transport of. It seems that the particular experimental setup is probably not cofactor-balanced, and how this weighs into the analysis needs to be made clearer earlier.

We reworked the enzymatic section, performed tests in variable ethanol and NAD⁺ concentrations and determined the K_m and K_{cat} values (lines 451-473). The results are now summarized in Table 3. We had to change our conclusions, since the kinetic parameters of encapsulated CutO are not lower than that of free CutO, but appear to be even higher, although not significantly. We also made the conclusions more temperate in considering the differences between native BMCs and our BDP particles and the limits and accuracy of our methods. Overall, this section contributes to a very general conclusion about the shell not being a major obstacle for CutO substrates. Still, we think this section has scientific merit and we would like to keep it in the manuscript. Our data are summarized in Table 3 and Supplementary Dataset 1.

3) As possibly relative newcomers the authors do a reasonable job overall with referencing, but there are some gaps that should be remedied.

- The first part of the introduction notes the range of BMC types. Jorda et al. 2013 (PMID 23188745) provided a bioinformatic articulation of the major BMC types that are now recognized.

Cited in line 42.

- In the second paragraph of the Introduction, pentameric vertex proteins are introduced. That discovery is due to Tanaka et al. in Science 2008, which should be credited.

Cited in line 60.

- The discovery that the main BMC shell proteins are hexamers is due to Kerfeld et al. in Science 2005.
Cited in line 55.

- The formation of flat sheets within crystals was noted in Kerfeld 2005, and the first demonstration of sheet formation outside of the crystal state is due to Dryden et al. 2009 (PMID 19844993).
Cited in line 55.

- The authors here use the name BMC-P for the vertex proteins. That protein family was first given the name BMV in 2013 (see Wheatley et al. PMID 23456886) in order to make clear that it is a different protein family from the BMC family. The BMC-P naming unfortunately loses that distinction. I suppose authors have leeway to choose naming conventions, but as a matter of priority if not clarity, it would be helpful to readers to make the equivalence clear by noting that that family is also known as BMV.
Cited in line 60 and the alternative name is now mentioned in parentheses.

- The third paragraph introduces the subject of glycy radical microcompartments. Jorda et al 2013 is where that important and diverse family was described. This paper has been overlooked in some recent papers on the glycy radical microcompartments; that should not be perpetuated here.
Cited in line 73.

Minor points:

- Given how vaguely defined the meaning of resolution is for EM, it probably makes more sense to keep the resolution value limited to one digit after the decimal, i.e. 3.3A.

We agree with the point and now keep the resolution value everywhere to one digit after the decimal as suggested.

- Is glycerol favored over glycerine as a common chemical name?

We stick to glycerol everywhere in the text. I am under the impression one can use both without any preference.

- There seems to be an extra sentence fragment at line 207.
Removed.

- On line 285, the authors describe the protein interactions as 'interlocking'. If they mean to stick to that description, exactly what does it mean?

We meant this as linking. We re-worded the sentence in line 430 to "These interactions result in a ring structure of four linked cmcC` monomers.". We hope it is now clearer.

- The authors discuss shell protein pore sizes (line 323 and elsewhere). They need to be more specific about what they are measuring as the pore size. Is this the distance between atom centers across the pore? Or have the van der Waals radii of the atoms been subtracted?

The van der Waals radii have been subtracted, this is now addressed in line 362.

- Figure 7 needs to be plotted a different way. As it is, the main observation is obscured.

Reworked and replaced by Table 3.

- The use of 6 significant digits for the B-factor (methods) is excessive, by multiple digits.

This was what the Relion program calculated and applied, with all the significant digits. We have rounded the value to -150 in the Supplemental table 2.

Reviewer #2 (Remarks to the Author):

This is potentially an important contribution to the study of the structure and function of bacterial microcompartments, which are coming to be recognized as a major factor in microbial metabolic diversity and because of their outstanding potential for use in bioengineering.

The authors focus on a previously uncharacterized type of organelle, the GRM2, which has significant implications for human health. The paper reports an impressive amount of data on both the delimiting shell and the internal organization and assembly of the enzymatic core of the GRM2 BMC (elucidation of the function of the different domains of the CutC enzyme in encapsulation is a significant contribution toward the understanding of GRM function, and BMC biogenesis in general), however some of the other conclusions should be reconsidered and tempered. If queries about aspects of the experimental work can be satisfactorily addressed and, overall, the findings could be better contextualized with the current understanding of BMCs, especially GRMs (a recent review by Ferlez et al (Mbio 2019) could provide a ready handle on this surprisingly abundant but poorly characterized family of BMCs that encapsulate glycol radical enzymes, including the different structural features of the GRE (cutC here) that seem likely related to assembly, this contribution to the field is certainly worthy of publication in Nature Communications.

Shell Characterization and Structure: This is a solid contribution about what can be formed from expressing different contributions of shell proteins outside of their native context and is done here in a completely new system. The authors should consider the pioneering work of Parsons et al., in this and also previous observations of nanotubes from shell proteins—this also relates to their shell particle classes.

Cited and discussed in discussion (lines 509-514).

Regarding the structure of the synthetic shell (needs to be clarified—this cannot be considered a proxy for the shell of an intact GRM2 metabolosome, much less an intact metabolosome (e.g. line 96 “minimum requirement for BMC formation; this is inaccurate—“BMC” should be reserved for shell + cargo, aka the native organelle. What has been obtained with cmcC+D is a minimal shell, it cannot even be considered a proxy for an empty native shell as it is missing the other shell proteins). Indeed, throughout the structure description it should be clear that this is an empty synthetic shell—likewise, calling it a particle can give the mis-impression that this is an intact BMC, or even a proxy shell containing all components; referring to “shell particles”, or making a definition statement earlier on will prevent misunderstanding.

This has been addressed by introducing a term “bacterial microcompartment shell derived particles” or BDPs to prevent any confusion with native BMC. Also, we now discuss the implications of our results in context of difference between native and recombinant shells in results (lines 110-113) and discussion (lines 490-496, lines 547-548 and lines 564-567).

Please provide electron density figures of the discussed features, e.g. Supp. Fig. 12/pores (also in Supp Fig. 12 the protein name of 2QW7 is CcmL, not cmcL). Indicate which residues are modelled / which are disordered/missing for the shell structure.

Electron density maps are now displayed in Supplementary Figure 24 and Supplementary Figure 21. Modeling data for cmcD is described in lines 375-377, for cmcC' in lines 340-342 and also in methods in lines 686-691. Mistake regarding CcmL naming in Supplementary Figure 21 has been corrected.

The model geometry should be better considering the high particle symmetry (high clashscore, 10% rotamer outliers, 2.2% Ramachandran outliers), maybe try refinement with phenix.refine. Please comment on the concave-convex properties of the shell protein, and the sidedness of the shell in the structure. This is an important comparison to a distinct type of BMC shell that has also been determined by cryo-EM (Greber, Sutter, Kerfeld 2019).

We did additional refinement in PHENIX, and the statistics is now much improved – clashscore is now 4.44, 0.4% poor rotamers, no Ramachandran outliers (Supplementary Table 2). Convex-concave properties are now commented in Figure 4b and lines 364-365.

In light of the cryo-EM structure, the question remains as to whether or not cmcA and cmcB are present in any of the GRM2 shells formed; would you obtain the same results (sizes, distributions, yields, etc.) if only cmcC' DE and were used to assemble shells? Do either cmcA or cmcB form homo-oligomeric structures that also have broad elution profiles on gel filtration that could be coeluting with the shells? The authors discuss and model the cmcC' hexamer and cmcD pentamer in the cryo-EM structure but do not mention the cmcA and cmcB hexamers included in the expression vector used to generate the 25 nm shells analyzed. Is it correct that the authors used the cmcABC' D + CutC336-1128 construct? If so, are cmcA and cmcB not present in these shells? This and the model quality (missing sidechains) and that the model is of insufficient resolution to distinguish paralogs should be made clear.

The observable electron density is most likely the average of all three BMC-H proteins, so all three BMC-H proteins are present. However, cmcC' is clearly the most abundant protein and the strongest contributor to electron density as suggested by the mass spectrometry analysis of cmcABC' +D (Supplementary figure 10a and 10b, now addressed in lines 336-340). This is how we justify the modeling of cmcC' in the position of BMC-H. We tested the assembly of cmcC' +D+E shell but we observed predominantly small type particles (Supplementary Figure 2c), so cmcA and cmcB are necessary for large type particle formation.

cmcA+D and cmcB+D did not form any observable assemblies in Superose 6 gel-filtration and cmcAB+D formed only predominantly small type BDPs (Supplementary Figure 2-3). Also, in the mass spectrometry study of cmcABC' +D BDPs we observed the presence of all three BMC-H proteins in more or less similar proportions irrespective of the size distribution of particles (Supplementary Figure 10). If the cmcA and cmcB are integrated in cmcABC' +D large type particles (not formed in cmcAB+D), it is very likely they are integrated in small type BDPs as well. Individual cmcA and cmcB assemblies coeluting with BDPs are very unlikely, since the interaction with other shell components would most likely prevent formation of such discrete formations made by a singular BMC-H protein. Model quality is discussed in lines 340-342, 375-377 and in methods (lines 687-692).

Considering the significance of the inter-hexamer gaps---this needs to be reconsidered in the context of the quality of the structure and, moreover, the non-native complement of shell proteins. Given its incompleteness in terms of structural building blocks, this is not a valid conclusion for a native shell. Likewise, the same incompleteness could potentially account for the failure to encapsulate certain

components. The Kerfeld group has suggested that the binding sites for EPs, for example, may be composed of sites formed by specific combinations/organizations of shell proteins.

We modified the claim and now refer strictly to BDPs without implying its importance in native BMC (lines 448-450). The incompleteness is only partial though – only 25% of T4 particles are missing the pentamers. Encapsulation process in the rest of the particles are unaffected by missing blocks, so it is unlikely that it affects the overall conclusions about encapsulation.

cmcB is not labeled in any main or supplementary figures. What is its expected molecular weight? Was it detected in shell containing gel filtration fractions? It is important to identify which of the expressed shell proteins are detected in each shell population as this will impact interpretation of encapsulation (e.g. inability to encapsulate CutH, or CutO/F in the absence of CutC) as well as structural properties of the resulting BMCs (e.g. does cmcA/B content effect size of the BMC shell?). Gel filtration results of individual shell proteins would also be useful to not only identify the same species in the gel filtration gels when other components are present but also detect any larger assemblies of a single species that may be co-migrating at the same size as assembled shells and so cause misinterpretation of what exactly is in the shell. Likewise, it would be useful to show the gel filtration profile of CutC alone to compare against the different elution patterns seen when coexpressed with the shell proteins.

cmcB is labeled in Figure 1 as part of cmcABC. We added molecular weights for all BMC structural and enzymatic components in Figure 1a as well.

We addressed the issue of the protein content of BDPs in a semi-quantitative approach by analyzing large type particle and small type particle gel-filtration fractions with mass spectrometry (lines 184-200, Supplementary figure 10). These data demonstrates that all components are present in the case of cmcABC`+D and, except for the amount of pentameric cmcD component, their proportions differs little.

Migration of individual BMC-H proteins would reveal nothing new about their interactions and possible subtypes of particles. It could be also a misinterpretation to make any conclusions about composite BDP particle content based on the migration of separate components in gel filtration – interaction may create new assembly properties, as was the case with separate cmcA+D and cmcB+D and cmcAB+D (Supplementary Figure 2b, 3c and 3d).

We performed several control experiments in gel filtration to prove that purified individual CutC, CutO and CutH proteins elutes differently and can be separated from BDPs (Supplementary Figures 17-18). CutF migration partially overlaps with BDP elution zone and it also binds to CutC, so we concede in lines 304-309 the possibility of CutC-CutF complexes being present in the case of cmcABC`+cmcD+CutC+CutO+CutF and cmcABC`+cmcD+CutC+CutF (Supplementary Figure 14).

Table1: There may be a typo here; the first two rows include two instances of the cmcABC + cmcD construct with different shell diameter distributions for each: the first states “predominantly 25 nm” and the second states “200-25 nm particles”. What is the difference between these two constructs, is one meant to be cmcABC`?

This is a typo, and you understood correctly – it was meant to be cmcABC`. The table is now corrected.

Line 120: Figure S1d does not show cmcABC + cmcD + cmcE (instead the gel is labeled as cmcABC + cmcD + CutC + cmcE)

Mistake corrected.

Fig. 2 What molecular weight do the peaks correspond to (compared to size standards; these should be presented), what is the measured absorption wavelength (A_{280} ?); later peaks seem low molecular weight but shells are observed in TEM.

There are limited choices for size markers in BDP range (above 2 MDa) for representation of BDPs. We chose bacteriophage AP205 virus like particles (VLPs) as the closest representative of BDPs available in our lab to mark their migration in gel filtration (Supplementary Figure 17). The AP205 VLPs migrate almost exactly as the small type particle peak, so the BDPs migrate in Superose 6 gel filtration as would be expected

In general, we do not measure and collect UV absorption spectra. We do not particularly trust these measurements because the amount of residual *E. coli* protein and DNA contamination can be very different among different runs. The UV spectrum of cmcABC`+D in Figure 2 is a best case scenario, but including spectra for all runs would add some very inconsistent data to an already extensive supplemental material. SDS-PAGE gels are much more reliable in this regard for analysis of the gel filtration process.

Figure 2a: please label location of BMC shell proteins in SDS-PAGE gel.

Labels provided.

Figure 2b: it would be valuable to provide histograms and average size calculations for the various size shells observed by TEM for each zone collected from the gel filtration column. This data could offer insight into assembly mechanisms.

Size distribution histograms were calculated from TEM micrographs with ImageJ software and are now displayed in Supplementary Figures 4-7.

Lines 106-107 and 128-129: The authors attribute the increase in yield of the cmcABC`D BMC shells, as compared to the cmcABCD shells, to the C-terminal extension on cmcC`; were the relative expression levels of cmcC and cmcC` examined? Could the increase in BMC shell formation be due simply to an increased cellular titer of cmcC` relative to cmcC as a result of the C-terminal extension? It's feasible to think that changes in the intracellular level of cmcC` could effect the stoichiometry of individual shell components such that the assembly of shells is favored and correspondingly more shells can be isolated during purification. To test the effect of expression levels of cmcC variants on shell assembly, the authors could use an inducible promoter for the expression of cmcC and show that shell yield and size does not vary as a function of induction (expression) levels.

It is certainly possible that cmcC` is more soluble, more available or more expressed than cmcC. However, to prove such impact on the assembly of cmcABC+D BDPs would require extensive efforts. Expression from separate promoters completely ignores the proximity effect of cmcABC protein synthesis in a polycistronic mRNA. Since it is possible that the introduction of additional protein via another promoter could not be comparable to expression in native cmcABC from single mRNA, it would require to do extensive research with ribosome binding site mutagenesis or introducing extra BMC-H gene copies in mRNA to be completely sure. This is too large scope for this study, but could be a good subject of future research.

We roughly compared the expression levels and solubility of cmcC` and cmcC in total cell lysates, but we did not observed any dramatic differences in expression between them (Supplementary

Figure 11). We did a crude effort to increase *cmcC* and *cmcAB* relative expression in *cmcABC*+*cmcD* BDPs by expressing the former two genes from additional promoter, but we failed to observe any major differences between *cmcABC*+*cmcD*+*cmcC* and *cmcABC*+*cmcD*+*cmcAB* gel filtration processes (Supplementary Figure 12, lines 210-216) – only perhaps slightly higher overall yield in case of additional *cmcC*.

As a negative control, was the *cmcC*_{trunc} variant used to assemble *cmcABC*_{trunc}*D* shells to compare yields with *cmcABCD* or *cmcABC*'*D* shells? The relative expression levels of *cmcC*, *cmcC*' , and *cmcC*_{trunc} should also be compared. The conclusion of 127-129 seems to be a bit of an overstatement.

We managed to make such a *cmcABC*_{trunc}+*D* construct and got some unexpected results – it turned out that truncation of *cmcC* in *cmcABC*_{trunc}+*D* BDPs also result in formation of larger type particles in a very similar manner as for *cmcC* in *cmcABC*' +*D* (lines 152-155).

We did some rough comparison of BMC-H protein expression levels and solubility by comparing total cell lysates (lines 201-209, Supplementary Figure 11). The expression levels are fairly similar, except for obviously lower levels for *cmcC*_{trunc}. Perhaps that is the reason why it is much more minor in composite *cmcABC*_{trunc}+*D*.

We removed this last claim since the results with *cmcABC*_{trunc}+*D* paints a more complex picture.

A general comment about the shell composition of GRM2 (line 69); this is a fairly average number of shell protein paralogs for a BMC.

We agree, corrected in lines 83-84.

Also, line 215-222: finding Hsp70 in an affinity purification of an over-expressed protein could also interpreted as adding to fold the over-expressed protein correctly. It is observed in overexpression of many distinct types of proteins (beyond BMCs). This observation is not particular to BMCs and this discussion should be removed.

We agree with your point and removed this section.

Enzymology: The conclusions of the activity assay results are very speculative, it might be better to just show a decrease in the activity when FeADH is encapsulated by itself encapsulated but this could be due to many factors, e.g. NADH/NAD⁺ permeability, improper assembly of the BMC cargo due to the lack of the other proteins and potential over-encapsulation of CutC which could act as an additional diffusion barrier. Isolation and investigation of one step in a pathway that should operate as a whole can be misleading. This section in general needs to be tempered in its conclusions.

Fig. 7 plotting specific activity against enzyme concentration reveals little about the potential interesting effects going on; maybe consider experiments with variable ethanol concentration.

We reworked the enzymatic section, performed tests in variable ethanol and NAD⁺ concentrations and determined the *K_m* and *K_{cat}* values (lines 452-473). We changed our conclusions, since the kinetic parameters of encapsulated CutO are not lower than that of free CutO, but even slightly higher. We also made the conclusions more temperate in considering the differences between native BMCs and our BDP particles and the limits and accuracy of our methods in results and discussion (lines 564-568). Overall, this section contributes to a rather general conclusion about the shell not being a major obstacle for CutO substrates. Still, we think this section has scientific merit and we would like to keep it in the manuscript. Our data are summarized in Table 3 and Supplementary Dataset 1.

Line 154: Although the authors cite previous results where a his tag was successfully used to isolate CutC, did the authors confirm via western blot the presence of the his-tag on CutC in the shell-containing gel filtration fractions to confirm it wasn't degraded during the shell isolation protocol?

We did this Western blot analysis as you suggested and confirmed the presence of the his6x tag on encapsulated CutC in Supplementary Figure 19c.

Lines 310-312: Were the 200 nm particles with CutC + CutF + CutO also assayed for CutO activity? Is the permeability of the larger shell any different than the smaller 25 nm shells shown? This would be interesting given the evidence that these small shells apparently contain only one hexamer (i.e. cmcC') based on the cryo-EM data.

The protein content, as demonstrated by mass spectrometry, differs little for large and small type cmcABC`+D BDP particles (with the exception of cmcD, being in lesser amounts in large type particles). Besides, considering the possibility of some encapsulated aggregates being present (lines 304-309), it would be hard to determine whether such aggregates are present and if yes, how they contribute to the total activity. Also, small type BDPs already have quite significant background activity and large type BDPs are even more impure and thus would likely have even more of this activity. So we chose not to test and present the enzymatic activity of such particles since we cannot be sure what exactly is measured.

The alcohol dehydrogenase characterization as presented seems to simply show a relationship between concentration and activity. Since this cofactor may be recycled within the lumen in the native BMC, did the authors investigate the permeability of choline using CutC activity as a readout?

Considering the complete insolubility of the CutD activating enzyme in our expression system, it is not possible to produce BDPs containing active CutC in our recombinant E. coli expression system.

Assembly: Genetic fusions of such sequences have been demonstrated to be sufficient for the encapsulation of nonnative proteins during propanediol utilization (Pdu)^{31-32, 34} and ethanolamine utilization (Eut)³⁵ in beta-carboxysome³⁶ BMC particles." Based on the references cited, I believe the authors are citing the evidence for EPs to tag nonnative cargo in these three types of shells (Pdu, Eut, and carboxysomes) separately; as written, it sounds like proteins from Pdu or Eut systems have been targeted to the carboxysome which I don't believe to be true.

This mistake has been corrected, lines 65-68.

Line 145: Authors state no encapsulation of the PTAC (CutH) was observed; were cmcE containing shells used as well as the cmcABC(D) shells reported? Could the authors comment on the EP found on the PTAC (see Erbilgin et al., Plos Biology for background on this). This also relates to the surprising observation that the Aldehyde dehydrogenase was not encapsulated, despite containing an EP. The authors should comment on this. Also, regarding the solubility of the activating enzyme; perhaps it needs to be co-expressed with the signature enzyme, as shown previously for a GRM3 BMC (Zarzycki et al. 2017). This is also offers an example of "piggy-backing" as the authors observe in their GRM2 system.

Large type shell cmcABC+D, full structural gene set cmcABC+D+cmcE shells and cmcABC`+D+CutC+CutH and cmcABC`+D+CutC+CutF+CutO+CutH constructs were tested (lines 222-224). We were actually quite surprised we failed to observe any comigration with BDPs. This could be due to the limits of our recombinant system as correct representatives of native BMCs. We address the

inactivity of EPs in results (lines 278-281) and discussion (lines 543-548) and cite this reference as well. We have actually tried to coexpress the CutD activating enzyme with CutC but it unfortunately had no effect on the solubility of CutD.

It is a bit of over-interpretation of these data to state “the size of the 25 nm particles is not effected by the presence or absence of cargo and is likely limited by the shell composition” —or that the core is predetermined by the shell size. Lines 184-206: authors conclude that final BMC size is “determined by the intrinsic properties of the shell proteins and not by the core” (lines 194-196); however, the inability of the core enzymes CutC, CutF, and CutO to effect the size of *cmcC*+D particles may be due to structural limitations of this pair of shell proteins to form a polyhedral compartment larger than 25 nm. Is the size distribution of *cmcABC*'D shell particles effected by the presence of the enzymatic core enzymes (CutC, CutF, CutO)? Perhaps with the extra shell components (i.e. *cmcA* and *cmcB*) the presence of the core enzymes can influence the size of the resulting shells. If the size distribution is unchanged this would support the authors' claim that the size of the particle determines the packaging of the core, rather than being limited by the shell components available.

We tempered our conclusions in this regard in discussion (lines 485-496). We now separate our conclusions about BDP shells from conclusions about native BMC shells. We did a crude effort to increase *cmcC* and *cmcAB* relative expression in *cmcABC*+*cmcD* BDPs by expressing the former two genes from additional promoter, but we failed to observe any major differences between *cmcABC*+*cmcD*+*cmcC* and *cmcABC*+*cmcD*+*cmcAB* gel filtration processes (Supplementary Figure 12, lines 210-216). We think the approach of using separate promoters ignores the native expression conditions in a polycistronic mRNA. It would be necessary to do additional experiments with mutating the RBS or adding extra copies of particular BMC proteins in mRNA sequence. This could be a good direction for future studies, but for an addition in this article it would be an excessive effort.

Other:

- please provide explanations of the used abbreviations, e.g. Cut, *cmc*—coordinating colors and symbols across all figures would be helpful to the reader, especially the non-BMC community.

Abbreviations are now given in page 3 and colors are now coordinated as much as possible – green for BMC-H, yellow for BMC-P and blue for core enzymes.

- please indicate molecular weights of the individual shell components and label corresponding bands on all SDS-PAGE gels

Molecular weights are given now in Figure 1 and bands are now labeled as much as possible in all gels.

- Fig. 1a color pentamer differently, and use consistent coloring throughout
Pentameric *cmcD* is now colored yellow everywhere.

- Fig. 4b consider recoloring, maybe use electrostatics representation to provide more information
We followed your advice and replaced these images with electrostatic potential maps.

- line 233 the particles of the circularly permuted HO BMC-H are pT=3 (not pT= 1)

No, in that particular case the shell particle consisted of 12 pentamers or 60 subunits (PMID:26988700), so it is a T=1 particle.

Reviewer #3 (Remarks to the Author):

This manuscript describes the characterization of several aspects of a GRM-type BMC. The most impactful aspect of the manuscript is the structural data on the particles that formed from various expression experiments. Figures 4 and 5 are all of high relevance to the field right now. Findings such as the structures of minor BMC structures (eg elongated) and the high resolution of the structure make it a valuable contribution to the field. Unfortunately, other aspects of the manuscript are lacking, as detailed below. In particular, data and controls are lacking for most of the other experiments, or conclusions are drawn from data that is not included in the figures, and strongly worded claims are not supported with references to the data.

General/Main Text:

- 1) Many conclusions are made about the null outcome of an experiment in the manuscript. For example, in Line 162: "CutO and CutF were not encapsulated..." The data does not show that CutO and CutF are not encapsulated, but rather that they were not detected. All such mis-statements should be corrected.
- 2) Similarly, language needs to be softened for claims that are not directly supported by the evidence. For example, Line 191-193, "It also seems that..." could be "One explanation for this result is that..." Or, "...the data suggest that the shell capacity..."

The language is now modified accordingly – we now generally claim only the observation of comigration and use the term "encapsulation" sparingly. We also rewrote our conclusions more critically; e.g. "When the CutC+CutF+CutO proteins were coexpressed with the cmcC`+D and cmcC+D constructs, capable of forming only small type BDPs, no comigration of any protein was detected, suggesting that the core size could be too large for encapsulation in small type particles".

- 3) No protein interaction studies are used to confirm hypotheses about the role of CutC. Such support is required to support the putative explanations; gel filtration assays are not sufficient. While the experiments described are important first steps, this section by itself is not well-developed enough for inclusion as a main finding without additional supporting evidence.

We tried to address these issues with additional control experiments. We performed several Superose 6 gel filtration control experiments to prove that purified individual CutC, CutO and CutH proteins elutes differently and can be separated from BDPs (Supplementary Figures 17-18). However, CutF migration partially overlaps with BDP elution zone and it also binds to CutC, so we concede in lines 304-309 the possibility of unencapsulated CutC-CutF complexes being present in the case of cmcABC`+cmcD+CutC+CutO+CutF and cmcABC`+cmcD+CutC+CutF (Supplementary Figure 14).

The data in Supplementary Figures 17-18 has some limited information about core protein-protein interaction. In our early studies we did some interaction studies by using his6-tagged CutC as a prey and untagged CutO as a target, and we did not observed any co-elution after coexpression of these enzymes. CutC-dependent interaction with CutO is nevertheless observable in the presence of BDPs (lines 263-270). The BDP system is not a perfect representation of native BMC, true, but interaction studies of separate components are even less reminiscent of native conditions. We think that the data about interaction and encapsulation of CutC and CutO are sufficiently convincing in light of additional control (Supplementary Figures 17-18; lines 240-263). In the case of CutF, we retreat from confirmation of encapsulation.

Figure 1:

1) The authors state that cmcA, B, and C are “highly similar”; a multiple sequence alignment, either in this figure or in the supplement would be useful for identifying potential critical residues for compartment assembly.

Alignment is now given in Figure 1. Alignment is also given in Supplementary Figure 8.

2) Line 110: “empty volume” is usually referred to as “void volume”

Corrected, in line 143.

Figure 2:

1) It would be useful if the authors would label the protein bands on the PAGE gel in 2A. Labels are present in the supplemental figures but not in the main text figure. The PAGE figures in the supplement should also have labeled molecular weight makers.

Bands are now labeled as much as possible in Figure 2A, since it is hard to distinguish separate proteins. All supplementary PAGE gel figures now have labeled molecular weight markers and labeled molecular weight bands.

2) Authors claim that particles of particular sizes elute in certain fractions but no sizing data is presented. They should include quantitative data, for example by either measuring the diameter of imaged particles or doing DLS on these purified fractions.

We evaluated particle sizes in TEM micrographs with ImageJ software (Supplementary Figures 4-7) and summarized the data in histograms displaying the size distribution.

Supplementary Figure 4d:

1) The authors claim that co-migration of CutC with BMC protein indicates encapsulation, but co-migration in SEC does not distinguish between luminal encapsulation, indirect association with the shell (via other co-eluting proteins, of which there are at least 20), direct binding to the shell exterior, or if it merely forms aggregates which are the same approximate molecular weight of the empty compartments (which vary dramatically from 25-200 nm in diameter). Any of these interactions when co-expressed could also shield effective purification using nickel affinity. The authors also do not show the HisTrap column elution profiles, which are central to their claims of encapsulation. The authors could effectively show encapsulation by fusing CutC to a fluorescent protein such as GFP and looking for fluorescent puncta in cells.

Supplementary Figure 5:

1) As with Supp Fig 4d, co-migration by SEC does not directly imply encapsulation, especially since the BMC shell proteins elute over such a broad range. Others have shown that BMC core enzymes natively aggregate. It's possible these enzymes aggregate into inclusion bodies that are in a size regime with a similar hydrodynamic radius to that of the empty BMCs. While these SEC results are promising, they are far from conclusive.

2) There are at least 20 other bands greater than 25 kD in MW, and the core enzymes make up a relatively small portion of the higher molecular weight bands present in the SEC profiles. This further supports their presence in nonspecific aggregates rather than inside of compartments.

3) The authors claim that the presence of CutF shifts encapsulation of the other enzymes towards the larger 200 nm compartments, but again, sizing data is absent. It is completely reasonable that inclusion of this protein increases aggregate size, rather than changing enzyme encapsulation preference for larger compartments.

We tried to address these issues with additional control experiments on Superose 6 gel filtration. We performed several Superose 6 gel filtration control experiments to prove that purified individual CutC, CutO and CutH proteins elutes differently and can be separated from BDPs (Supplementary Figures 17-18).

To check whether there is some unspecific binding between BDPs and core enzymes, we expressed CutC+CutO and CutC+CutO+CutF core enzymes and cmcABC`+BDPs in separate biomass batches, then mixed them together and purified as usual (Supplementary Figure 18b-e). If there would be a significant direct or E. coli protein-mediated binding between BDPs and core enzymes, the comigration should be observable in this mixing setup as well. CutC+CutO in such a mixed setup were not present – for comigration they have to be coexpressed, and this further confirms the encapsulation (Supplementary Figure 18c). However, there is indeed some validity about unspecific binding of the larger CutC+CutF+CutO aggregates. CutF migration partially overlaps with BDP elution zones and it also binds to CutC even further increasing the aggregation (Figure 17); also, it is possible to detect the presence of CutC in CutC+CutF+CutO biomass mixture with BDPs (Supplementary Figure 18e), so we concede in lines 302-307 the possibility of unencapsulated CutC-CutF complexes being present in the case of cmcABC`+cmcD+CutC+CutO+CutF and cmcABC`+cmcD+CutC+CutF (Supplementary Figure 14). We also admit that in the case of CutC+CutF+CutO it is not possible to confirm encapsulation (lines 304-309).

There are no elution profiles available for HisTrap experiments in Supplementary Figure 19 since these experiments were done by hand and syringe. We consider PAGE analysis of elution and flowthrough fractions and control to be sufficient for proving our point, since all samples were equalized by volume and loaded on the gel in equalized volumes as well (now mentioned in methods, lines 616-617).

We made the CutC-GFP fusion construct and did some experiments. We tested the solubility of CutC-GFP, and it was soluble in comparable amounts with CutC. CutC-GFP also comigrated with cmcABC`+D BDPs in gel filtration just as CutC. However, we observed fluorescent points both in the case of CutC-GFP and also in the case of cmcABC`+D+CutC-GFP. Unfortunately, it was not possible to do any major conclusions about encapsulation. It could be that CutC is prone to forming either aggregates or inclusion bodies in E. coli cells. We added the obtained fluorescence pictures below.

CutC-GFP

cmcABC`+D+CutC-GFP

Supplementary Figure 10:

1) Remove or even further soften claims. It is pure speculation that co-purification of an E. coli chaperone with a heterologously over-expressed unstructured BMC protein from K. pneumoniae implies that chaperones are necessary for compartment formation. It is much more likely that overexpressing an unstructured protein recruits chaperones.

We agree with your point and removed this section.

Supplementary Figure 13:

1) The authors could not find any electron density indicative of encapsulated enzymes, perhaps because the enzymes are not actually encapsulated. This possibility is not considered, yet seems the most obvious.

In some BDPs something is actually encapsulated, as we did identify an asymmetric C1 subclass containing some sort of electron density located in the lumen (Supplementary Figure 22). However, we cannot positively identify a blob at 20-30 angstrom resolution as CutC (lines 393-402). Since we are convinced that CutC and CutO are encapsulated, we can still reasonably suspect CutC₃₃₆₋₁₁₂₈ as this object inside. This particular finding does not conclusively confirm or deny anything.

Figure 7:

1) The authors do not indicate whether there is a significant difference in activity between encapsulated and unencapsulated protein. There are also more points for unencapsulated than encapsulated enzyme, which is not explained.

2) There is no empty BMC control. There is also a surprising amount of activity at low enzyme concentration, possibly indicative of some baseline contamination.

We reworked the enzymatic section, performed tests in variable ethanol and NAD⁺ concentrations and determined the K_m and K_{cat} values (lines 452-473). The results are now summarized in Table 3. We had to change our conclusions, since the kinetic parameters of encapsulated CutO are not lower than that of free CutO, but slightly higher instead. We also made the conclusions more temperate in considering the differences between native BMCs and our BDP particles and the limits and accuracy of our methods. Overall, this section contributes to a rather general conclusion about the shell not being a major obstacle for CutO substrates (at least not in orders of magnitudes). Still, we think this section has some scientific merit and we would like to keep it in the manuscript. Our data are summarized in Table 3 and Supplementary Dataset 1.

There is now an empty BDP control. There is some residual contamination left in empty BDPs, and this is mentioned now in lines 459-461.

REVIEWERS' COMMENTS:

Reviewer #1 (Remarks to the Author):

The paper has been substantially improved by editing. There are also important changes to the results on shell protein permeability. It's a concern that the authors apparently had it wrong in their original work, and it's not so clear what happened in the new studies to give very different results. But the new findings are at least more in line with expectation (especially for an artificial shell with probable defects of missing pieces).

Reviewer #2 (Remarks to the Author):

The authors have done an acceptable job of addressing my concerns.

Reviewer #3 (Remarks to the Author):

The revised manuscript is much improved. I have some minor edits to suggest, and one more comment.

1) BMC is still used instead of BDP in a few places, eg lines 100, 149, 157, 229, 328, 329, 342, 348, 349, 404, 439, 646, 651, 652, and perhaps more. It doesn't seem that any of the particles studied here are native BMCs with all cargo and wild type shell proteins, so I expect all of these will need to be changed.

2) Per the methods, the enzyme activity results are based on 2 replicates of a single biological sample. Is there evidence that the variability will be not with the purification but with the enzyme assay itself? It is more standard to show 3 or more biological replicates for such assays if so. Related to enzyme assays, I would like a scheme in the supplemental section that shows which enzymes are present, what their reactions are, etc, to help me (and future readers) evaluate the relevance of the presented enzyme activity data as measured by NAD/NADH turnover. That being said, I am confident that the data represent the experimental results, but I am not convinced that the experiment is designed to measure what is stated.

REVIEWERS' COMMENTS:

Reviewer #1 (Remarks to the Author):

The paper has been substantially improved by editing. There are also important changes to the results on shell protein permeability. It's a concern that the authors apparently had it wrong in their original work, and it's not so clear what happened in the new studies to give very different results. But the new findings are at least more in line with expectation (especially for an artificial shell with probable defects of missing pieces).

Eventually, it was decided to remove entire enzymatic assay section due to data reliability issues such as these, so these concerns are no longer relevant. It is true that this relatively minor part of the article was the least trustworthy.

Reviewer #2 (Remarks to the Author):

The authors have done an acceptable job of addressing my concerns.

Reviewer #3 (Remarks to the Author):

The revised manuscript is much improved. I have some minor edits to suggest, and one more comment.

1) BMC is still used instead of BDP in a few places, eg lines 100, 149, 157, 229, 328, 329, 342, 348, 349, 404, 439, 646, 651, 652, and perhaps more. It doesn't seem that any of the particles studied here are native BMCs with all cargo and wild type shell proteins, so I expect all of these will need to be changed.

We corrected this and changed BMC to BDP in all necessary places.

2) Per the methods, the enzyme activity results are based on 2 replicates of a single biological sample. Is there evidence that the variability will be not with the purification but with the enzyme assay itself? It is more standard to show 3 or more biological replicates for such assays if so. Related to enzyme assays, I would like a scheme in the supplemental section that shows which enzymes are present, what their reactions are, etc, to help me (and future readers) evaluate the relevance of the presented enzyme activity data as measured by NAD/NADH turnover. That being said, I am confident that the data represent the experimental results, but I am not convinced that the experiment is designed to measure what is stated.

Eventually, it was decided to remove entire enzymatic assay section due to data reliability issues such as these, so these concerns are no longer relevant. It is true that this relatively minor part of the article was the least trustworthy.